# EXPERT expands prime editing efficiency and range of large fragment edits

Youcai Xiong[1,2,8], Yinyu Su[1,2,8], Ruigao He[1,2,8], Xiaosong Han[1,2,3], Sheng Li[1,2], Minghuan Liu[1,2], Xiaoning Xi[1,2], Zijia Liu[1,2], Heng Wang [1,2], Shengsong Xie [1,2,4,5], Xuewen Xu[1,2,4,5], Kui Li [6], Jifeng Zhang [7], Jie Xu [7], Xinyun Li [1,2,4,5] ✉, Shuhong Zhao [1,2,3,4,5] ✉ & Jinxue Ruan [1,2] ✉

Prime editing systems (PEs) hold great promise in modern biotechnology. However, their editing range is limited as PEs can only modify the downstream sequences of the pegRNA nick. Here, we report the development of the extended prime editor system (EXPERT) to overcome this limitation by using an extended pegRNA (ext-pegRNA) with modified 3' extension, and an additional sgRNA (ups-sgRNA) targeting the upstream region of the ext-pegRNA. We demonstrate that EXPERT can efficiently perform editing on both sides of the ext-pegRNA nick, a task that is unattainable by canonical PEs. EXPERT exhibits prominent capacity in replacing sequences up to 88 base pairs and inserting sequences up to 100 base pairs within the upstream region of the ext-pegRNA nick. Compared to canonical PEs such as PE2, the utilization of the EXPERT strategy significantly enhances the editing efficiency for large fragment edits with an average improvement of 3.12-fold, up to 122.1 times higher. Safety wise, the use of ups-sgRNA does not increase the rates of undesirable insertions and deletions (indels), as the two nicks are on the same strand. Moreover, we do not observe increased off-target editing rates genome-wide. Our work introduces EXPERT as a PE tool with significant potential in life sciences.

The initial prime editing (PE) system was constructed by fusing a Strep. pyogenes Cas9 (SpyCas9) nickase (H840A) with an engineered Moloney-Murine leukemia virus reverse transcriptase (M-MLV RT)[1]. To enable editing, a prime editing guide RNA (pegRNA) is used, which includes a protospacer defining the target site, a sgRNA scaffold, and a 3' extension encoding the desired edit. This extension contains a primer binding site (PBS) complementary to a segment of the DNA protospacer, as well as an RT template (RTT) that encodes the desired edit

and genomic homologous sequences[1]. PEs hold great promise in modern biotechnology. However, all existing PEs can only modify the downstream genomic region of the pegRNA nick, but cannot modify the upstream genomic region, posing a significant limitation on their editing range. In addition, the editing efficiency of PEs remains to be improved.

There have been efforts to expand the editing range of PE. In one study, substituting SpyCas9 nickase (H840A) with Francisella novicida

[1]Frontiers Science Center for Animal Breeding and Sustainable Production, Huazhong Agricultural University, Wuhan, PR China. [2]Breeding and Reproduction of Ministry of Education & Key Laboratory of Swine Genetics and Breeding of Ministry of Agriculture and Rural Affairs, Huazhong Agricultural University, Wuhan, PR China. [3]Yazhouwan National Laboratory, Sanya, PR China. [4]The Cooperative Innovation Center for Sustainable Pig Production, Huazhong Agricultural University, Wuhan, PR China. [5]Hubei Hongshan Laboratory, Huazhong Agricultural University, Wuhan, PR China. [6]Agricultural Genomics Institute at Shenzhen, Chinese Academy of Agricultural Sciences, Shenzhen, PR China. [7]Center for Advanced Models for Translational Sciences and Therapeutics, University of Michigan Medical School, Ann Arbor, MI, USA. [8]These authors contributed equally: Youcai Xiong, Yinyu Su, Ruigao He. ✉e-mail: xyli@mail.hzau.edu.cn; shzhao@mail.hzau.edu.cn; ruanjinxue@mail.hzau.edu.cn

Cas9 nickase (H969A) expands the editing range of PE[2]. The circular RNA-mediated prime editor (CPE) which combines Cas12a with circular RNA[3], and template-jumping prime editing (TJ-PE)[4], inspired by the genomic insertion mechanism of retrotransposons, also expand the editing range. The performance of PE has also been enhanced through other strategies. For example, adding motifs, such as modified prequeosine₁-1 riboswitch aptamer (evopreQ1)[5], Csy4[6], G-quadruplexes[7], viral exoribonuclease-resistant RNA motif (xrRNA)[8], or MS9[9], to the 3′ extension of the pegRNA improves the stability of the pegRNA and consequently the PE efficiency. Protein engineering to PE system, such as altering the composition of nuclear localization signals[10], introducing point mutations to alter certain amino acid residue[11] or adding additional peptide or protein domains (e.g., IGFpm1-NFATC2IPp1-PE2 (IN-PE2)[12], chromatin-modulating peptide-PE-Variant1 (CMP-PE-V1)[13], hyPE2[14], PE combined with the recruitment of P65 protein[15]), as well as adding the dominant negative MLH1 (MLH1dn) mutant to inhibit the mismatch repair pathway, is also effective in improving PE efficiency[11]. Nevertheless, while many PE systems could achieve double-digit (i.e., >10%) efficiencies in human cells, they can only do so within approximately a 10 bp range downstream of the nick[1,11]. Hence there is a need for further expanding the editable region of PE systems, particularly for editing sites located more than 10 bp away downstream from the nick, and editing sites located upstream from the nick.

Recently, the use of two-pegRNA, such as in dual-pegRNA[16], twinPE[17], GRAND (genome editing by RTTs partially aligned to each other but nonhomologous to target sequences within duo pegRNA)[18], PRIME-Del[19], homologous 3′ extension mediated prime editor (HOPE)[20], and bi-direction prime editing (Bi-PE)[21], has gained momentum in expanding the editing range of PE and improving its efficiency. It should be noted however, that in these two-pegRNA systems the two nicks are located on the complementary strands, i.e., in *trans*, which is associated with an elevated probability of inducing double-strand breaks (DSBs)[22,23]. Furthermore, the editable region is positioned in between the downstream direction of these two-pegRNA nicks. In other words, these two-pegRNA approaches are still unable to edit the upstream region of either pegRNA nick.

Here, we report the development of the EXPERT. EXPERT uses an additional sgRNA (ups-sgRNA) that is located upstream of pegRNA to generate a *cis* nick of the pegRNA nick, and an extended pegRNA (ext-pegRNA) that has an elongated and modified 3′ extension. We compared EXPERT with the representative single-pegRNA system PE2 and the two-pegRNA system twinPE, demonstrating that EXPERT achieves higher product purity. The results reveal that EXPERT can efficiently perform editing on both sides of the ext-pegRNA nick, a task that is unattainable by canonical PEs, while maintaining low indel rates at the target site and minimal off-target effects genome-wide. Additionally, the EXPERT strategy proves particularly useful for large fragment edits. Collectively, EXPERT represents an effective gene-editing tool that complements the PE toolbox and holds significant value in life sciences.

## Results
### EXPERT strategy
Figure 1a illustrates the design of EXPERT. The EXPERT, in comparison to the canonical PEs, has two differences: (1) it has an additional ups-sgRNA that targets the upstream genomic region of the pegRNA nick. As a result, EXPERT generates two nicks on the same strand, which refer to as "*cis* nicks". (2) it has a modified pegRNA, designated as ext-pegRNA, which has an elongated and modified 3′ extension. The ext-pegRNA comprises a PBS and a reverse transcriptase template (RTT). The RTT is composed of an edit sequence (ES) and a homologous sequence (HS) (Supplementary Fig. 1). The PBS on the 3′ end of the ext-pegRNA binds to the DNA strand of the 3′ Flap, which is generated by the ups-sgRNA

(Fig. 1a). To distinguish from the binding of canonical pegRNA, we refer to this binding as "upstream binding".

Both *cis* nicks and upstream binding are essential for the differential editing capacity of the EXPERT compared to canonical PEs. We first confirmed the role of the two *cis* nicks in EXPERT, by comparing it with two variants, EXPERT-a and EXPERT-b, which each generate only one nick using either the ups-sgRNA or the ext-pegRNA, respectively (Supplementary Fig. 2a). These variants were then used to edit the *HEK4_2* locus and introduce a 40-bp sequence replacement in HEK293T cells (Supplementary Fig. 2b). As expected, EXPERT (with two *cis* nicks) achieved 6.1% efficiency, in contrast to the 2.84% and below 0.1% efficiencies achieved by EXPERT-a and EXPERT-b, respectively. Mechanistically, we speculate that this is because EXPERT generates two *cis* nicks that enhance the detachment of the original single-stranded DNA fragment from the genome, thereby promoting subsequent processes (Supplementary Fig. 2c). These results confirmed the importance of the two *cis* nicks for the EXPERT.

We next aimed to confirm the role of the upstream binding. We constructed several PE2 variants (Supplementary Fig. 2a). (i) PE2: it generates one nick by using the ups-sgRNA as its pegRNA. It does not have the upstream binding. (ii) PE2-a: it generates one nick by using the ups-sgRNA as its pegRNA. It does not have the upstream binding. We included a truncated ext-pegRNA in PE2-a although it does not create a 2^nd nick. (iii) PE2-b: it generates two *cis* nicks, one using the ups-sgRNA, and another by using the ext-pegRNA. PE2-b also does not have the upstream binding. We compared these three PE2 variants with EXPERT, which has the upstream binding, to edit the same *HEK4_2* locus for introducing a 40-bp sequence replacement (Supplementary Fig. 2b). All these PE2 variants lacking the upstream binding design, regardless of the design to generate one nick or two *cis* nicks, achieved low efficient edits, ranging from 0.05% to 0.37%, consistent with the knowledge that current PEs are inefficient for large fragment edits. Remarkably, the efficiency achieved by EXPERT is 6.1%, 122.1-fold higher than that by PE2. These results confirmed the essential role of the upstream binding for the EXPERT.

One consideration with the generation of two nicks in EXPERT is whether this will increase the unintended indel rates at the on-target site. Our results in the above experiments suggest that the presence of two *cis* nicks does not increase the likelihood of indel events in comparison to the one nick system PE2. The indel rate at the *HEK4_2* site for EXPERT was 0.28%, comparable to that by PE2 (with 0.2% indels) (Supplementary Fig. 2b).

In summary, by introducing an extra upstream guide RNA (ups-sgRNA) to create an additional *cis* nick and an extended pegRNA (ext-pegRNA) that contains an upstream binding sequence, we construct a PE tool EXPERT.

### EXPERT expands the editing range allowing precise editing on both sides of the pegRNA nick with a minimal indel rate
Canonical PEs are unable to edit the upstream region of the pegRNA nick. The EXPERT, thanks to the introduction of two *cis* nicks and the use of ext-pegRNA, in theory, should be able to edit that unreachable region by canonical PEs.

To validate this hypothesis, we performed two edits at the *VEGFA_1* site, both located in the upstream region of the ext-pegRNA nick: (i) replacing a 37-bp sequence (*VEGFA_1* -37to-1 replace 37 bp); (ii) deleting a 37-bp sequence (*VEGFA_1* -37to-1 del 37 bp) (Fig. 1b). The *VEGFA_1* site was chosen because it provides multiple NGG sequences on the same strand, thereby facilitating the design of ext-pegRNA and ups-sgRNA. For the first edit, a high replacement efficiency at 33.7% with low indel rate (0.52%) was achieved. Similarly, for the second edit, the precise deletion rate was high at 18.8% with a low indel rate at 0.8%. We also attempted a third edit at this site: (iii) *VEGFA_1* -37to-1 replace extended loxp and +1 TtoC, to test whether EXPERT allows for simultaneous editing on both sides of the ext-pegRNA nick. The results

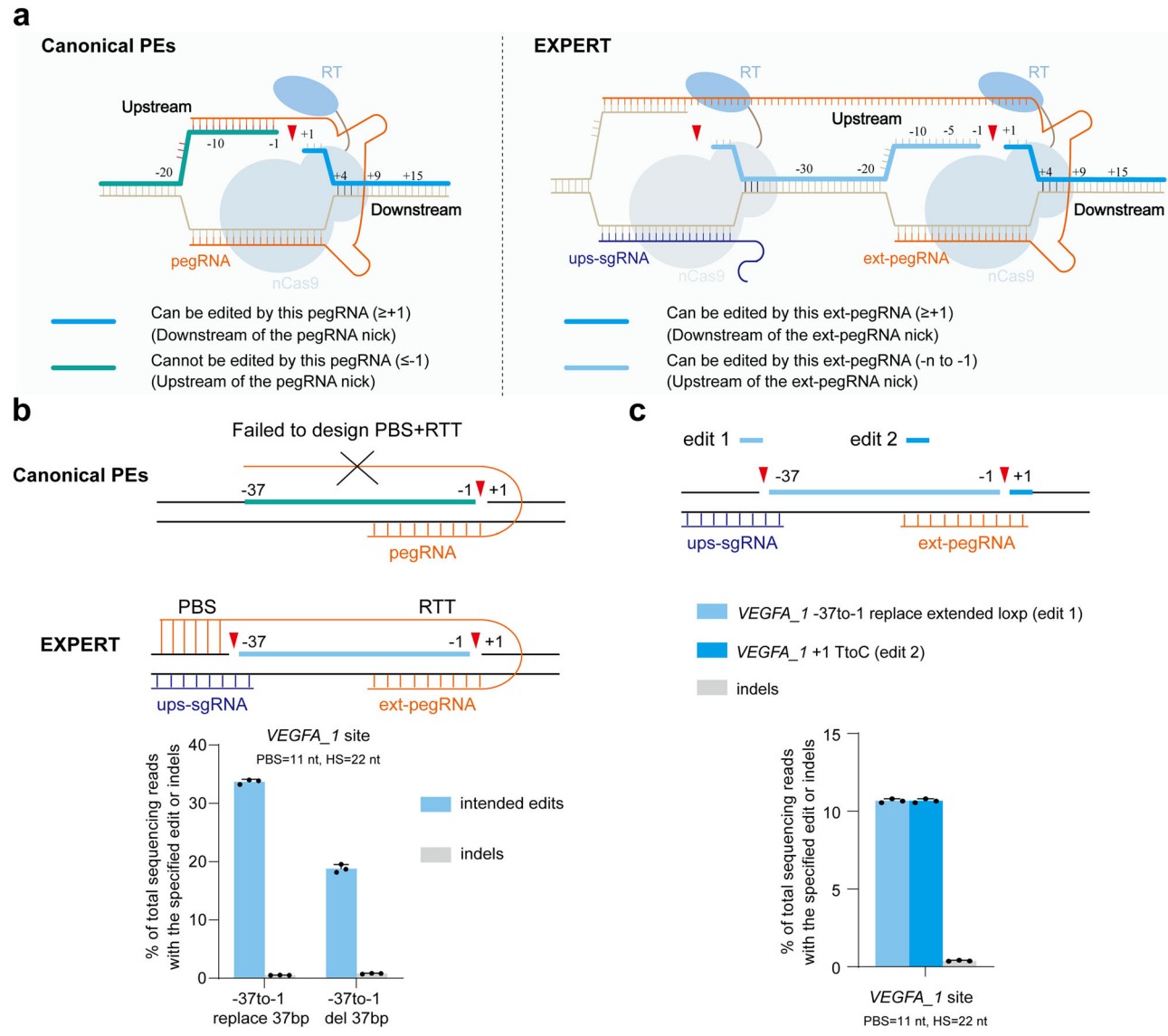

**Fig. 1 | EXPERT expands the editing range of canonical PEs. a** Schematics of canonical PEs and EXPERT. The deep blue thick-lined area (downstream region of the pegRNA nick) represents the editable region for canonical PEs and EXPERT. On the other hand, the green thick-lined area (upstream region of the pegRNA nick) is uneditable by canonical PEs, whereas the region marked by the light blue thick-lined area can be edited by EXPERT. nCas9, Cas9 nickase (H840A); RT, reverse transcriptase. **b** Frequencies of intended edits and indels introduced by EXPERT for two different edits, which canonical PEs cannot perform. PBS, primer binding site. RTT, reverse transcriptase template. Bars represent the mean of $n = 3$ independent biological replicates. Data are presented as mean ± s.d. **c** EXPERT performs simultaneous editing on both sides regions of the pegRNA nick at the *VEGFA_1* site. Bars represent the mean of $n = 3$ independent biological replicates. Data are presented as mean ± s.d. All sequencing data were collected from transfection-positive cells. Source data are provided as a Source Data file.

showed that the efficiency of this simultaneous editing on both sides of the ext-pegRNA nick has reached 10.7%, with a low indel rate at 0.38% (Fig. 1c).

These results demonstrate that EXPERT has the unprecedented ability to edit upstream region of the pegRNA nick, and simultaneously edit both sides regions. All types of edits have low indel rates, which preliminarily supports the hypothesis that the presence of two *cis* nicks does not elevate the occurrence of indel events.

### Effects of the distance between the two *cis* nicks on the editing efficiency of EXPERT

The distance between the two *cis* nicks (DCN) generated by ups-sgRNA and ext-pegRNA is a critical parameter in EXPERT. To evaluate the impact of DCN on editing efficiency, we utilized a 293T-reporter cell line containing a premature TAG stop codon in the coding sequence of mCherry[10], and tested EXPERT with DCNs of varying sizes, ranging

from 23 to 126 nt. A successful editing on the stop codon sequence will restore the expression of mCherry signal, which is used as a proxy of successful editing (Fig. 2a and Supplementary Fig. 3). The results showed that the mCherry signals were detectable when the DCNs ranged from 38 to 96 nt, with more robust signals observed when DCNs were between 38 and 71 nt, and editing efficiencies ranging from 1.74% to 7.44%.

We then assessed the impact of DCN on editing efficiency by EXPERT at two endogenous loci: *VEGFA_1* and *HEK4_1*. The results showed that the editing rates peaked when the DCNs were between 32 and 40 nt (Fig. 2b). At the *VEGFA_1* locus, the highest editing rate of 33.7% with an indel rate of 0.52% was achieved when the DCN was 37 nt long. At the *HEK4_1* locus, the highest editing rate of 11.9% with an indel rate of 2.23% was achieved when the DCN was 32 nt long. Interestingly, at this site (*HEK4_1*), a slightly shorter DCN of 24 nt almost completely abolished the intended edits. Similar results were observed in the

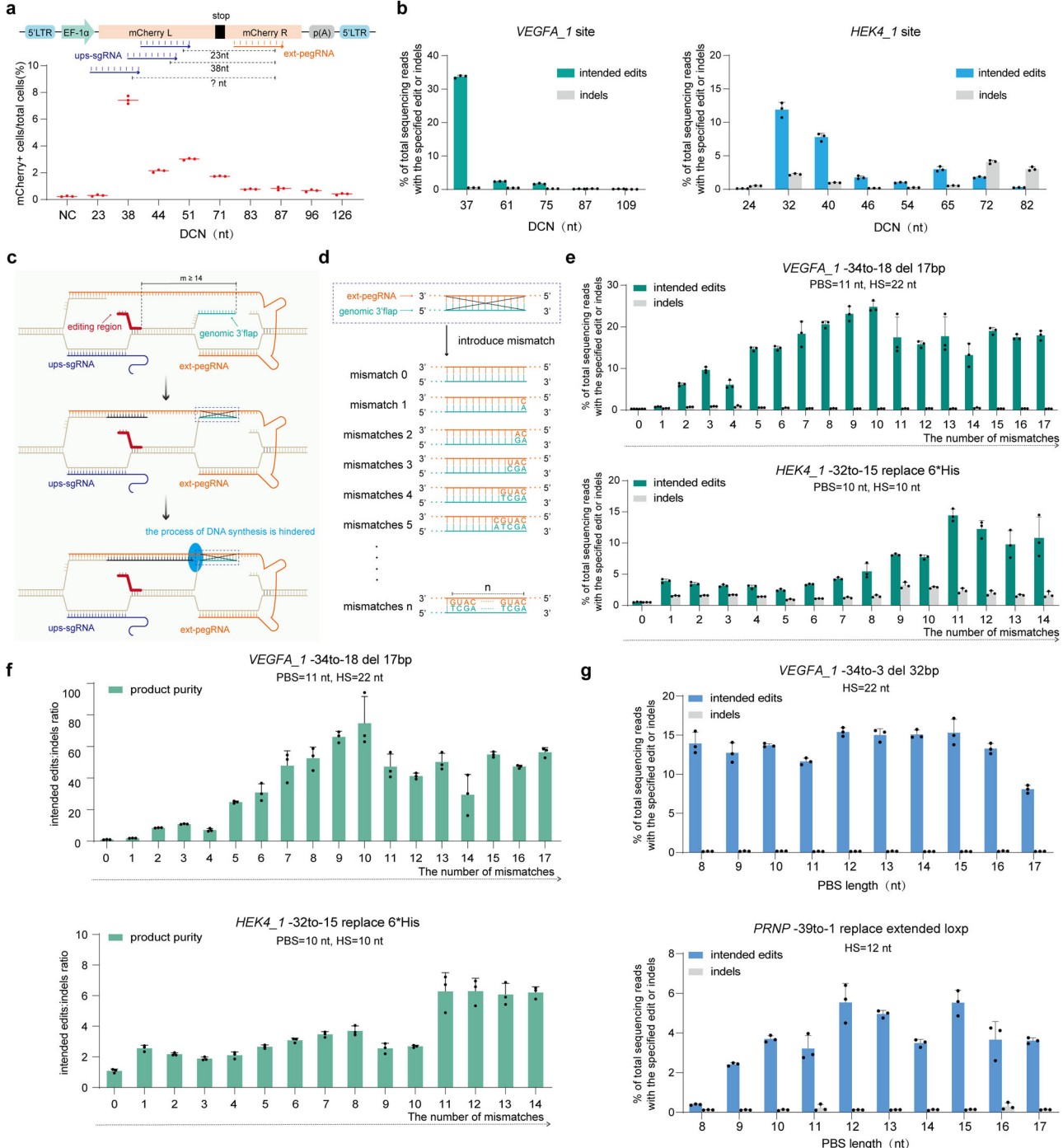

**Fig. 2 | Enhancing EXPERT editing efficiency through optimizing the distance between ups-sgRNA and ext-pegRNA, introducing mismatches on ext-pegRNA, or optimizing PBS length. a** Schematic diagram illustrating the modification of the stop codon in the 293T-reporter cell line to restore mCherry function under DCNs of varying sizes (top). Frequencies of edits introduced by EXPERT under DCNs of varying sizes were quantified using flow cytometry (bottom). Bars represent the mean of $n = 3$ independent biological replicates. **b** Frequencies of intended edits and indels introduced by EXPERT under DCNs of varying sizes were quantified at *VEGFA_1* and *HEK4_1* sites. Bars represent the mean of $n = 3$ independent biological replicates. Data are presented as mean ± s.d. **c** Diagram of the hybridization of the 3′ Flap of the original DNA strand to complementary ext-pegRNA. The hybridization can hinder the reverse transcription process of RT enzymes, leading to editing failure. The bold red line area denotes the editing region. "m" indicates the distance

between the editing region and ext-pegRNA nick. **d** The pattern of introducing mismatch on ext-pegRNA region that hybridizes with the 3′ Flap of the original DNA strand. **e** Frequencies of intended edits and indels introduced by EXPERT under different numbers of mismatches on the ext-pegRNA region that hybridizes with the 3′ Flap. Bars represent the mean of $n = 3$ independent biological replicates. Data are presented as mean ± s.d. **f** The product purity (intended edits: indels ratio) introduced by EXPERT under different numbers of mismatches on the ext-pegRNA region. Bars represent the mean of $n = 3$ independent biological replicates. Data are presented as mean ± s.d. **g** The impact of PBS length on editing efficiency of EXPERT. Bars represent the mean of $n = 3$ independent biological replicates. Data are presented as mean ± s.d. All sequencing data were collected from transfection-positive cells. Source data are provided as a Source Data file.

293T-reporter cells: when a 23 nt DCN was used, the editing rate was only 0.24%, whereas, with a 38 nt, the editing rate peaked at 7.44% (Fig. 2a). We speculate that these observations suggest that a DCN of 24 nt or shorter may cause steric hindrance between two adjacent nCas9 proteins. Based on this, DCNs of 24 nt or shorter should be avoided in EXPERT.

These data demonstrate that the distance between two nicks affects the editing efficiency of EXPERT. Our results suggest using a DCN between 32 and 40 nt as the starting design.

### Introducing mismatches on the ext-pegRNA improves the editing efficiency of EXPERT

When the editing region is away from either of the nick sites (i.e., the ext-pegRNA nick and the ups-sgRNA nick), there is a possibility for the genomic 3′ or 5′ Flap of the original single-stranded DNA to hybridize with the complementary ext-pegRNA, which could potentially hinder the synthesis by the reverse transcriptase and subsequently affect the production of the newly synthesized DNA strand (Fig. 2c and Supplementary Fig. 4a). In support of this, EXPERT exhibited notably low editing activity when the editing regions were away (≥14 nt) from the ext-pegRNA nick, both at the *VEGFA_1* site (edit: *VEGFA_1* -34to-18 del 17 bp, 0.3%) and the *HEK4_1* site (edit: *HEK4_1* -32to-15 replace 6*His, 0.54%) (Fig. 2e, mismatch number = 0). Similarly, when the editing region was away (≥14 nt) from the ups-sgRNA nick, EXPERT also showed low editing activity at the *HEK4_1* site (edit: *HEK4_1* -18to-1 replace 6*His, 0.28%) (Supplementary Fig. 4c, mismatch number = 0). Likely due to the same reason, EXPERT also demonstrated poor performance in creating small edits in the upstream region of the ext-pegRNA nick (Supplementary Fig. 5a).

We thus hypothesized that introducing mismatches in the regions corresponding to DNA 3′ Flap complementarity on the ext-pegRNA will disrupt the hybridization of DNA Flap and ext-pegRNA, therefore improving the editing efficiency of EXPERT.

We first tested if this strategy improves the EXPERT's editing efficiency to the editing regions that are away from the ext-pegRNA nick. We introduced mismatches starting from the nucleotide on the ext-pegRNA corresponding to the first nucleotide of the complementary DNA 3′ Flap, and tested the effects of the number of mismatches from 0 to 17 (Fig. 2d and Supplementary Fig. 6). The results demonstrated that when mismatches were introduced, as few as only one, the editing efficiency could be improved, for example at the *HEK4_1* site from 0.54% to 3.93%. The extent of improvement was positively correlated with the number of mismatches introduced. The highest editing efficiency was achieved with 10 or 11 mismatches at both tested sites: 24.81% at the *VEGFA_1* site and 14.43% at *HEK4_1* site (Fig. 2e). Analysis of product purity (the ratio of intended edits: indels) revealed that introducing mismatches enhanced product purity at both the *VEGFA_1* and *HEK4_1* sites (Fig. 2f).

We then tested if this strategy improves the EXPERT's editing efficiency to the editing regions that are away from the ups-sgRNA nick. Following a similar strategy, we introduced mismatches starting from the nucleotide on the ext-pegRNA corresponding to the first nucleotide of the complementary DNA 5′ Flap, with the number of mismatches varying from 0 to 14 as a way to improve EXPERT's editing efficiency to the editing regions that are away from the ups-sgRNA nick (Supplementary Fig. 4 and Supplementary Fig. 6). The results again demonstrated a positive correlation between the editing rates and the number of mismatches. When three mismatches were introduced, the editing efficiency reached ~10%. When 11 mismatches were introduced, the efficiency increased to 12.4%, and under these conditions, the highest product purity was achieved (Supplementary Fig. 4c).

We also tested an alternative strategy for introducing mismatches, specifically by introducing mismatches at intervals of every 3 and 5 nucleotides, respectively (Supplementary Fig. 7). The results showed that the editing efficiency of ext-pegRNA with such

mismatches closely resembles that observed with ext-pegRNA containing 11 consecutive mismatches. These results suggest an alternative mismatch design for ext-pegRNA to achieve effective editing with EXPERT. Moreover, EXPERT exhibits differential unintended indel rates at the *VEGFA_1* and *HEK4_1* loci, possibly attributable to the sequence characteristics of these sites or the editing sequence. It is important to note that incorporating appropriate mismatches can enhance editing efficiency and reduce indel rates (Fig. 2e and Supplementary Fig. 4c).

Lastly, we evaluated EXPERT for introducing small edits (small insertions, small deletions and base substitutions) in the upstream region of the ext-pegRNA nick, with or without the inclusion of mismatches (Supplementary Fig. 5). The results demonstrated that EXPERT can perform all these types of small edits in the upstream region of the ext-pegRNA nick (Supplementary Fig. 5). Again, introducing mismatches significantly improved editing efficiency, in line with our speculation that the introduced mismatches reduce the homology between the ext-pegRNA and the genomic DNA 3′ Flap, thereby preventing their hybridization (Fig. 2c). Nevertheless, it is noted that, despite the lower unintended on-target indel rates, the efficiency and product purity of the small edits achieved by EXPERT are generally lower than those achieved by PE3 (Supplementary Fig. 5b).

All these results demonstrate that the editing efficiency of EXPERT can be improved by introducing mismatches.

### Optimization of PBS length and HS Length in EXPERT

PBS plays an important role in initiating reverse transcription and ensuring efficient PE efficiency[1,24]. To assess whether the length of the PBS affects the editing efficiency of EXPERT, experiments were conducted at two loci: the *VEGFA_1* and *PRNP* loci, with PBS lengths ranging from 8 to 17 nt (Fig. 2g). The results showed that PBS lengths ranging from 9 to 16 nt could lead to efficient editing outcomes: 11.64% to 15.41% at the *VEGFA_1* locus, and 2.42% to 5.53% at the *PRNP* locus. From a practical point, we would recommend using PBS length of 12 nt because at both loci this parameter led to the highest editing efficiency, and avoid using PBS shorter than 9 nt as apparently this drastically reduced the editing rate in one tested locus.

Subsequently, we conducted an investigation to determine the optimal length of HS. Analysis of the editing efficiency across different HS lengths revealed that effective editing could be achieved with HS lengths ranging from 12 to 22 nt (Supplementary Fig. 8). When the HS length is shorter than 16 nt, only one locus exhibits an editing efficiency greater than 6% (1 out of 12 or 8%). In contrast, when the HS length is 16 nt or longer, six loci show an editing efficiency exceeding 6% (6 out of 7 or 86%) (Supplementary Fig. 8). Therefore, we recommend an HS length of 16 nt or longer as the starting point in EXPERT design.

### Use of a Helper gRNA further improves EXPERT for the insertion and replacement of large DNA fragments at the upstream region of the ext-pegRNA nick

Next, we investigated the capacity of EXPERT to insert DNA fragments of different sizes (5, 10, 20, 30, 50, 80, 100 bp) at the upstream region of the ext-pegRNA nick (Fig. 3a, b). We tested these two loci: *EMX1* and *VEGFA_1*. In both sites, greater than 10% editing rates were achieved when the lengths of the insertion fragment were 50 bp. However, it is clear that the rates sharply dropped when the insertion size was large (e.g., 80 and 100 bp), although greater than 5% insertion rates of the 100 bp fragment were still achieved at both sites (Fig. 3b).

These results indicate that EXPERT is capable of inserting large DNA fragments (up to 100 bp) at the upstream region of the ext-pegRNA nick, but the efficiency remains to be improved. To address this, we included an extra gRNA, positioned between the ups-sgRNA and ext-pegRNA, which we refer to as the Helper gRNA (Fig. 3c). We reason that this Helper gRNA would direct the nCas9-RT to create a third *cis* nick, and recruit an additional nCas9-RT enzyme between the

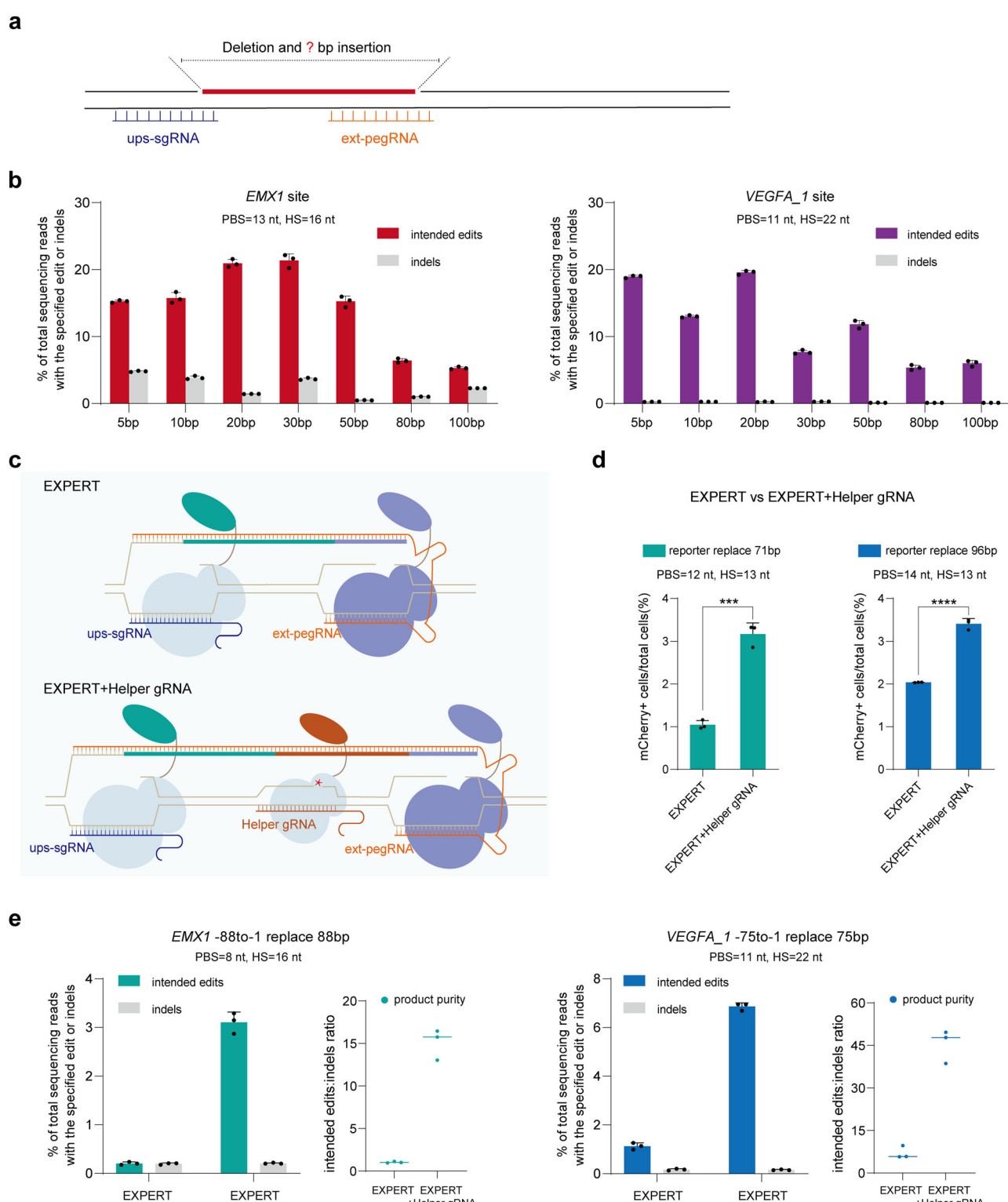

**Fig. 3 | EXPERT enables the insertion and replacement of large DNA fragments at the upstream region of the ext-pegRNA nick. a** Schematic diagram for deletion of the sequence between two nicks and insertion of fragments of different lengths. **b** Frequencies of fragment insertions of varying lengths (5, 10, 20, 30, 50, 80, 100 bp) and indels introduced by EXPERT at the *EMX1* and *VEGFA_1* sites, respectively. Bars represent the mean of *n* = 3 independent biological replicates. Data are presented as mean ± s.d. **c** Schematics of EXPERT and EXPERT + Helper gRNA. In comparison to EXPERT, EXPERT + Helper gRNA incorporates an additional sgRNA in the middle region between the ups-sgRNA and ext-pegRNA, referred to as the Helper gRNA. **d** Frequencies of edits introduced by EXPERT and EXPERT + Helper gRNA were quantified using flow cytometry. Bars represent the mean of *n* = 3 independent biological replicates. Data are presented as mean ± s.d. The *P* value was calculated using a two-tailed *t*-test, and no adjustments were made for multiple comparisons. $P_{\text{reporter replace 71 bp}} = 0.0002$, $P_{\text{reporter replace 96 bp}} = 0.00005$; ***$P < 0.001$; ****$P < 0.0001$. **e** Frequencies of intended edits, indels, and product purity introduced by EXPERT and EXPERT + Helper gRNA. Bars represent the mean of *n* = 3 independent biological replicates. Data are presented as mean ± s.d. All sequencing data were collected from transfection-positive cells. Source data are provided as a Source Data file.

original two nicks, thereby promoting the detachment of the original DNA strand and increasing the overall reverse transcription.

We tested this design in the 293T-reporter cell line. The addition of the Helper gRNA to the system resulted in a 3-fold improvement for a 71-bp replacement edit and a 1.7-fold improvement for a 96-bp replacement edit (Fig. 3d). This was further confirmed at two endogenous sites: (i) an 88-bp replacement at the *EMX1* site; and (ii) a 75-bp replacement at the *VEGFA_1* site (Fig. 3e). The replacement efficiency at the *EMX1* locus increased from 0.21% to 3.1%, a 15-fold improvement. Likewise, at the *VEGFA_1* locus, the efficiency increased 6.1-fold from 1.13% to 6.86%. In all cases, the indel rates were at low levels ranging from 0.15% to 0.2%, further indicating that introducing nicks on the same DNA strand does not elevate the risk of generating DSBs.

These findings demonstrate that including a Helper gRNA in the EXPERT improves its capacity for large fragment replacement editing without elevating the indel rate.

## EXPERT enhances the product purity of prime editing for large fragments

The use of the ups-sgRNA in the EXPERT raises the question of whether it introduces higher indel rates. To systematically evaluate this, we compared EXPERT with the representative single-pegRNA system PE2 and the two-pegRNA system twinPE.

**Comparison of EXPERT with PE2.** We first compared the editing outcome and the indel rates between EXPERT and the single-pegRNA system PE2, both of which induce nicks on a single DNA strand. The ups-sgRNA was used as the pegRNA for PE2. We conducted comparative experiments between EXPERT and PE2 in 19 different edits (including insertions, deletions, and replacements) at nine loci. Detailed information of specific edits is shown in Fig. 4a and supplementary Fig. 9.

The results demonstrated an overall improvement in the editing efficiency by EXPERT over that of PE2 when editing large fragments (Fig. 4a and Supplementary Fig. 9). For insertion edits spanning from 24 to 66 bp, EXPERT achieved 1.5 to 3.4-fold higher editing efficiency than PE2 (Supplementary Fig. 10). For deletions edits spanning from 30 to 43 bp, the improvement by EXPERT over PE2 ranged from 1.7 to 15.7-fold (Supplementary Fig. 10). For equidistant base replacements edits ranging from 34 to 43 bp, the improvement by EXPERT over PE2 is between 1.3 to 3.2-fold (Supplementary Fig. 10). The overall performance, counting these types of edits in all loci, is 1.3 to 15.7-fold better by EXPERT than that by PE2, with an average improvement of 3.12-fold (Fig. 4b). No differences of on-target indel rates were observed between EXPERT and PE2 edited cells (Fig. 4c).

The above results indicate that the EXPERT strategy enhances the editing efficiency of PE2 for large fragment edits without increasing the on-target indel rates.

**Comparison of EXPERT with twinPE.** We next compared EXPERT with one two-pegRNA system twinPE. EXPERT induces both nicks on the same DNA strand (i.e., *cis* nicks), whereas the twinPE induces one nick on each strand (referred to as *trans* nicks). We speculate that *trans* nicks, in comparison to *cis* nicks, increase the likelihood of DNA DSBs and subsequent indels. To validate this speculation, we conducted comparative experiments between EXPERT and twinPE at four different loci. Briefly, they are: (i) *VEGFA* 36 bp deletion and 78 bp insertion, (ii) *LMNA* 36 bp replacement, (iii) *VEGFA_1* 37 bp deletion and 103 bp insertion, (iv) *HEK4* 37 bp deletion and 79 bp insertion. The twinPE pegRNA pairs used for each locus were selected after a pre-screening process (Supplementary Fig. 11).

Our findings reveal varied but comparable editing efficiencies between these two systems: EXPERT showed higher efficiency at the *VEGFA_1* locus, whereas twinPE outperformed EXPERT at the *LMNA* and *HEK4* loci, and both systems exhibited similar performance at the

*VEGFA* locus (Fig. 4d). Importantly, purity analysis demonstrated that EXPERT achieved higher purity in all edits, with increases ranging from 1.5 to 5.7-fold compared to twinPE (Fig. 4d). It should be noted that although a pre-screening process for twinPE pegRNA pairs has been conducted, a more thorough screening (including additional pegRNA pairs, varying PBS lengths, and different combinations) will be essential in the future to enable a more comprehensive comparison of the performance between EXPERT and twinPE.

We also want to point out that unlike the two-pegRNA systems which require two separate PAM sequences (NGG) on both DNA strands, the EXPERT only needs PAM sequences on one strand, which allows EXPERT to target additional regions. Such comparison experiment cannot be conducted because these regions are only targetable by EXPERT but not by twinPE or any two-pegRNA systems.

## Low genomic off-target events by EXPERT

Next, we evaluated the gRNA (ext-pegRNA or ups-sgRNA)-independent and gRNA-dependent off-target (OT) effects of EXPERT using whole-genome sequencing (WGS). The EXPERT-expressing plasmids carrying three types of targeting RNAs were used: (i) with non-target gRNAs as the vehicle control; (ii) targeting the *EMX1* locus (*EMX1* -17 loxp + 6 * His + AscI ins); and (iii) targeting the *VEGFA_1* locus (*VEGFA_1* -34to-3 del 32 bp). The plasmids were transfected into HEK293T cells, respectively. The genomic DNA of transfected cells of these groups, as well as those of the non-transfected wild-type cells, were subjected to WGS.

We first determined whether EXPERT induces gRNA-independent OT mutations genome-wide. The analysis showed that there was no significant difference in the level of genome-wide base substitutions and indels among the four groups of cells, indicating that EXPERT does not induce whole-genome gRNA-independent OT effects (Fig. 4e, f).

Next, we evaluated the gRNA-dependent OT effects of EXPERT. We determined the on-target editing efficiencies of EXPERT with amplicon sequencing and assessed the frequencies of indels in potential OT sites by WGS. The results showed that the frequencies of indels for all predicted potential OT sites are below 0.1% (Supplementary Fig. 12).

These results demonstrate that both gRNA-independent and gRNA-dependent OT effects by EXPERT are minimal.

## The EXPERT strategy is compatible with different PE systems

In theory, the use of ups-gRNA and ext-pegRNA is compatible with all PE systems as long as they are based on nicking the target DNA.

To demonstrate this, we constructed EXPERTmax by introducing ups-sgRNA and ext-pegRNA to the PE2max (Fig. 5a), and conducted edits at different loci. Detailed information of edit type is shown in Fig. 5a. Comparing to PE2max, EXPERTmax resulted in an average of 3.7-fold higher editing efficiency in all loci tested (Fig. 5a and Supplementary Fig. 13).

We further compared EXPERTmax with later versions of PEmax (PE3max, PE4max, and PE5max). Comparing to PE2max, PE3max uses an additional nicking sgRNA, PE4max uses an additional MLH1dn component, whereas PE5max utilizes both additional components (i.e., nicking sgRNA and MLH1dn).

Without including the additional nicking sgRNA, EXPERTmax had an average 1.7-fold higher editing efficiency than PE3max (Supplementary Fig. 14). Inclusion of the additional nicking sgRNA slightly but not dramatically further improved the EXPERTmax's editing efficiency, resulting in an average 1.9-fold higher than those by PE3max. EXPERTmax + MLH1dn outperformed PE4max by an average of 3.6-fold, and EXPERTmax + MLH1dn + additional nicking sgRNA resulted in an average 2.4-fold increase in editing efficiency compared to PE5max (Fig. 5a and Supplementary Fig. 13). Furthermore, product purity results demonstrated that EXPERTmax consistently achieved higher purity compared to its corresponding PEmax systems (Supplementary Fig. 15).

These results demonstrate that the EXPERT strategy is compatible and readily adaptable to other PE systems.

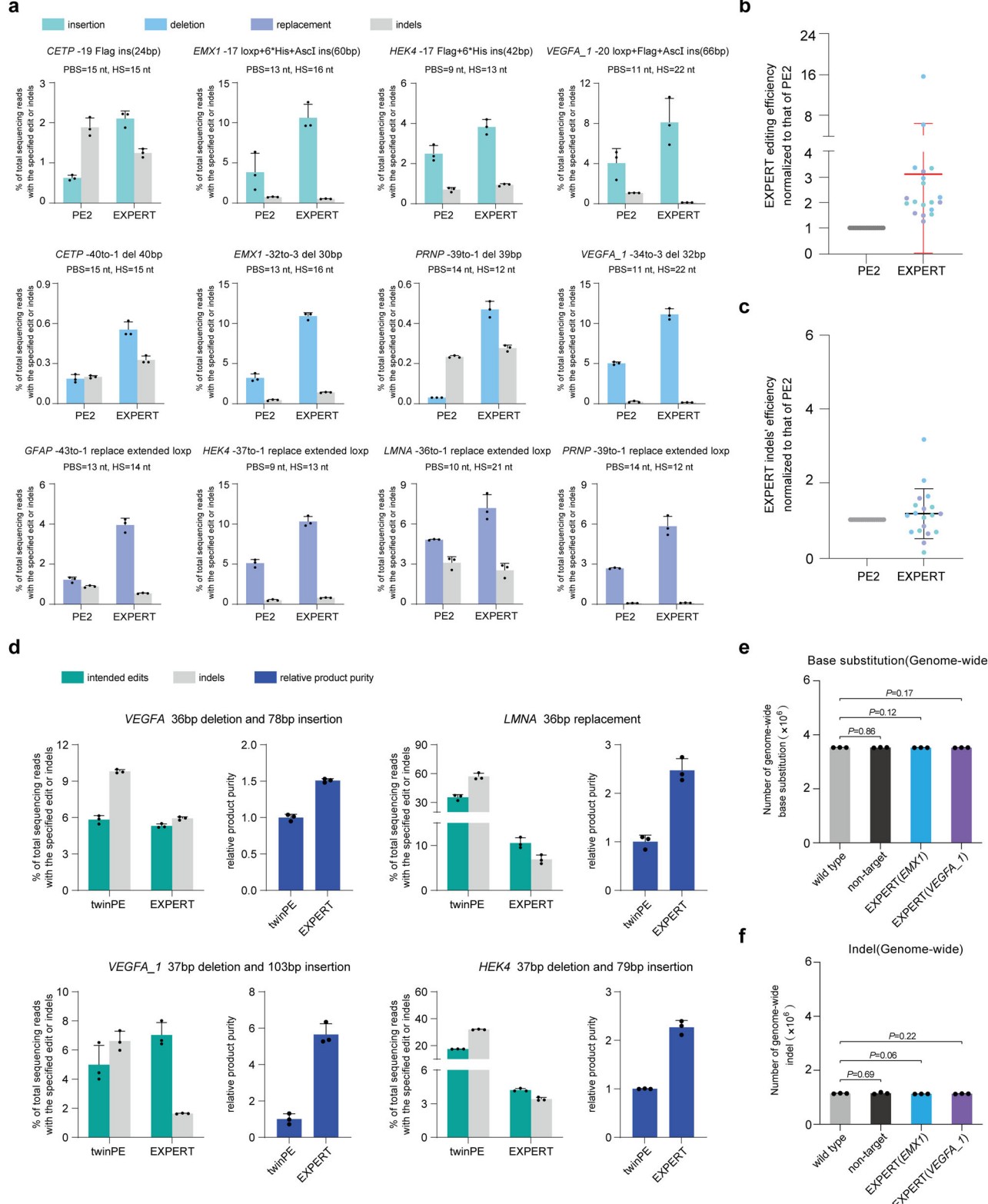

## Efficient prime editing by EXPERTmax in different cell types from multiple species

We next assessed EXPERTmax in different cell types in different species.

First, we used EXPERTmax to edit four different human cell lines: the lymphoblast cell line K562, the leukemia T lymphocyte cell line Jurkat, the cervical carcinoma-derived cell line Hela, and the human fetal lung fibroblast HFL1, respectively.

In K562 cells, the editing efficiency of EXPERTmax was 9.6–21.6 times higher than that of PE2max. The average improvement was 15.4-fold (Fig. 5b and Supplementary Fig. 16a). For instance, the efficiency of inserting a Flag tag at the *EXM1* site (*EMX1* -17 Flag ins) increased from 2.07% to 25.27%. In Jurkat cells, the editing efficiency of EXPERTmax was 2.9–46 times higher than that of PE2max. The average improvement was 17.4-fold (Fig. 5c and Supplementary Fig. 16b). The efficiency of inserting a Flag tag at the *EXM1* site (*EMX1* -17 Flag ins) increased

**Fig. 4 | The EXPERT strategy enhances editing efficiency with high product purity and low off-target effects. a** Frequencies of intended edits and indels introduced by PE2 and EXPERT at multiple loci. Additional mismatches were introduced in the insertion-type edits. Bars represent the mean of *n* = 3 independent biological replicates. Data are presented as mean ± s.d. **b** Statistical analysis of normalized editing frequencies, setting the frequencies induced by PE2 as 1. *n* = 19 editing from independent experiments shown in (**a**) and supplementary Fig. 9. Data are presented as mean ± s.d. **c** Statistical analysis of normalized frequencies of indels, setting the frequencies induced by PE2 as 1. *n* = 19 editing from independent experiments shown in (**a**) and supplementary Fig. 9. Data are presented as mean ± s.d. **d** Frequencies of intended edits, indels, and relative product purity introduced by twinPE and EXPERT at different loci. Bars represent the mean of *n* = 3 independent biological replicates. Data are presented as mean ± s.d. **e** Numbers of genome-wide base substitutions. Bars represent the mean of *n* = 3 independent biological replicates. Data are presented as mean ± s.d. The *P* value was calculated using a two-tailed *t*-test, and no adjustments were made for multiple comparisons. **f** Numbers of genome-wide indels. Bars represent the mean of *n* = 3 independent biological replicates. Data are presented as mean ± s.d. The *P* value was calculated using a two-tailed *t*-test, and no adjustments were made for multiple comparisons. All sequencing data were collected from transfection-positive cells. Source data are provided as a Source Data file.

from 0.07% to 3.22%, representing a remarkable 46-fold increase. Similarly in Hela cells, the efficiency by EXPERTmax was 1.5 to 8.4-fold higher than that by PE2max, with an average improvement of 4-fold (Fig. 5d and Supplementary Fig. 16c). In addition, the overall product purity of EXPERTmax was higher than that of PE2max in these three cell lines (Supplementary Fig. 17). In HFL1 cells, the efficiency of EXPERTmax was 1.5 to 25.8-fold higher than that of PE2max, while the unintended indel rates remained either lower than or comparable to those of PE2max (Supplementary Fig. 18). These findings demonstrate that the EXPERT can be utilized in different human cell types.

We then tested EXPERTmax in a mouse cell line N2a and in pig fetal fibroblast (PFF) cells to assess its applicability in non-human species. In N2a cells, the intended edits are "-34to-1 replace loxp" at the *Ifnar1* locus, and "-32to-1 replace Flag+AscI" at the *Tgfb1* locus. In pig PFF cells, the intended edits are "-17 Flag ins" at the *ANTXR1* locus, and "-41to-1 replace anti-JEV mutation" at the *CALR* locus. The results demonstrated that EXPERTmax significantly enhanced editing efficiency in both mouse and pig cells. Compared to PE2max, the editing efficiency of EXPERTmax was 3.9–4.3 times higher in N2a cells (Fig. 5e) and 3.4–39.7 times higher in PFF cells (Fig. 5f). Of note, in PPF cells at the *susANTXR1* site, the editing efficiency increased from 0.23% to 9.13%, representing a 39.7-fold increase. Consistent with findings in HEK293T cells, the unintended indel rates did not significantly increase compared to PE2max for most edits. As a result, the overall product purity achieved by EXPERTmax was also higher than that by PE2max (Fig. 5e, f).

These findings suggest that EXPERTmax can be used in different mammalian species. Notably, in pig species, there have been no previous reports of using PE systems for large fragment edits (e.g., replacement of 41 bp), as attempted here. EXPERTmax achieved an extraordinary efficiency of 23.4%.

### Generating disease-relevant mutations by EXPERTmax

We then employed EXPERTmax to generate disease-relevant mutations in human cells as a proof-of-concept demonstration for its potential use in human biomedical research. Cystic fibrosis (CF) is a fatal inherited disease caused by mutations in the cystic fibrosis transmembrane conductance regulator (*CFTR*) gene, which is inherited in an autosomal recessive manner[25–27]. To date, over 1900 mutations have been identified, with more than 300 known to cause the disease[28]. Traditional gene-editing techniques often have limited capabilities, leading to suboptimal therapeutic outcomes. To demonstrate EXPERT's applicability in CF gene editing, here we worked to generate a large fragment mutant sequence (36 bp) in exon 4 that carries multiple known *CFTR* mutations (Fig. 5g). This region cannot be edited by canonical PEs using the initially selected pegRNA because it is located at the upstream region of this pegRNA. However, it can be achieved using EXPERT by using this pegRNA spacer. We compared the editing efficiencies among (i) PE2max (using the ups-sgRNA as the pegRNA); (ii) PE3max (using the ups-sgRNA as the pegRNA); (iii) EXPERTmax (using the initially selected pegRNA as the ext-pegRNA); and EXPERTmax (using the initially selected pegRNA as the ext-pegRNA) + an additional nicking sgRNA (as used in PE3max) (Fig. 5g).

The results demonstrated that the utilization of EXPERTmax (9.18%) significantly enhanced the editing efficiency in comparison to PE2max (2.33%) and PE3max (3.66%). The use of "EXPERTmax + nicking sgRNA" further enhanced the efficiency over that by EXPERTmax along, reaching an average editing efficiency of 11.8%, 3.2 times higher than that by PE3max and 5.1 times higher than that by the PE2max. Moreover, EXPERTmax exhibited remarkably low indel rates and showed higher product purity compared to PEs (Fig. 5h). These findings suggest the potential value of EXPERTmax in translational research of human diseases such as CF.

## Discussion

In the present work, we report the development of a PE tool EXPERT. The EXPERT, comparing to canonical PEs, uses an extra ups-sgRNA, which generates an additional nick that is *cis* to the one generated by the ext-pegRNA. The 3′ extension of the ext-pegRNA binds to the region targeted by the ups-sgRNA. Our data suggest that the presence of an additional *cis* nick and the upstream binding of the 3′ extension of the ext-pegRNA enables efficient editing at the upstream region of the ext-pegRNA nick. As such, EXPERT expands the editing range, in particular including the upstream region of the ext-pegRNA nick, which is not attainable by all existing other PEs. Efficiency-wise, the EXPERT strategy significantly enhances the efficiency for large fragment edits, showing an average improvement of 3.12-fold (up to 122.1 times higher) compared to PE2. Safety-wise, the EXPERT strategy does not increase the risks of off-target effects and generally results in comparable on-target indel generation relative to PE2.

In the EXPERT, the two nicks are *cis*, i.e., on the same strand. Our data suggest that *cis* nicks, at least when their distances are below 88 nt, do not increase the risks of DSBs as compared to single nick. In fact, the EXPERT-associated indel rates observed in the "PE2 *vs* EXPERT" experiments are in the range from 0.1% to 4.2%, comparable to or lower than those generated by PE2. This is even true when a third *cis* nick is added in the "EXPERT + Helper gRNA" design. These findings are consistent with a recent report that paired *cis* nicks, up to 366 nt apart as tested in that work, scarcely create indels at the edited genomic loci[29].

Several other PE systems, such as twinPE and other two-pegRNA systems, also generate more than one nick. Of note, in these systems nicks are introduced to both DNA strands, i.e., in *trans*, which is known to induce DSBs[22,23] and other destructive events[30–32]. Indeed, the indel rate was as high as 57% by twinPE, in sharp contrast to the low indel rates (ranging from 1.6% to 6.9%) by EXPERT as observed in the "twinPE *vs* EXPERT" experiment. Our data thus suggest introducing *cis*, but not *trans*, nicks as an effective and safe means to improve the performance of the PEs.

The single-pegRNA-based PE systems are relatively inefficient in generating the large fragment insertions or replacements. As demonstrated in the present work, with the help of one or two additional *cis* nicks, the EXPERT is capable to efficiently insert or replace large fragments (75–100 bp). Mechanistically, we reason that this is because nCas9-RTs were recruited to each of nick sites to work collaboratively in a relay manner to complete the large fragment synthesis. It is

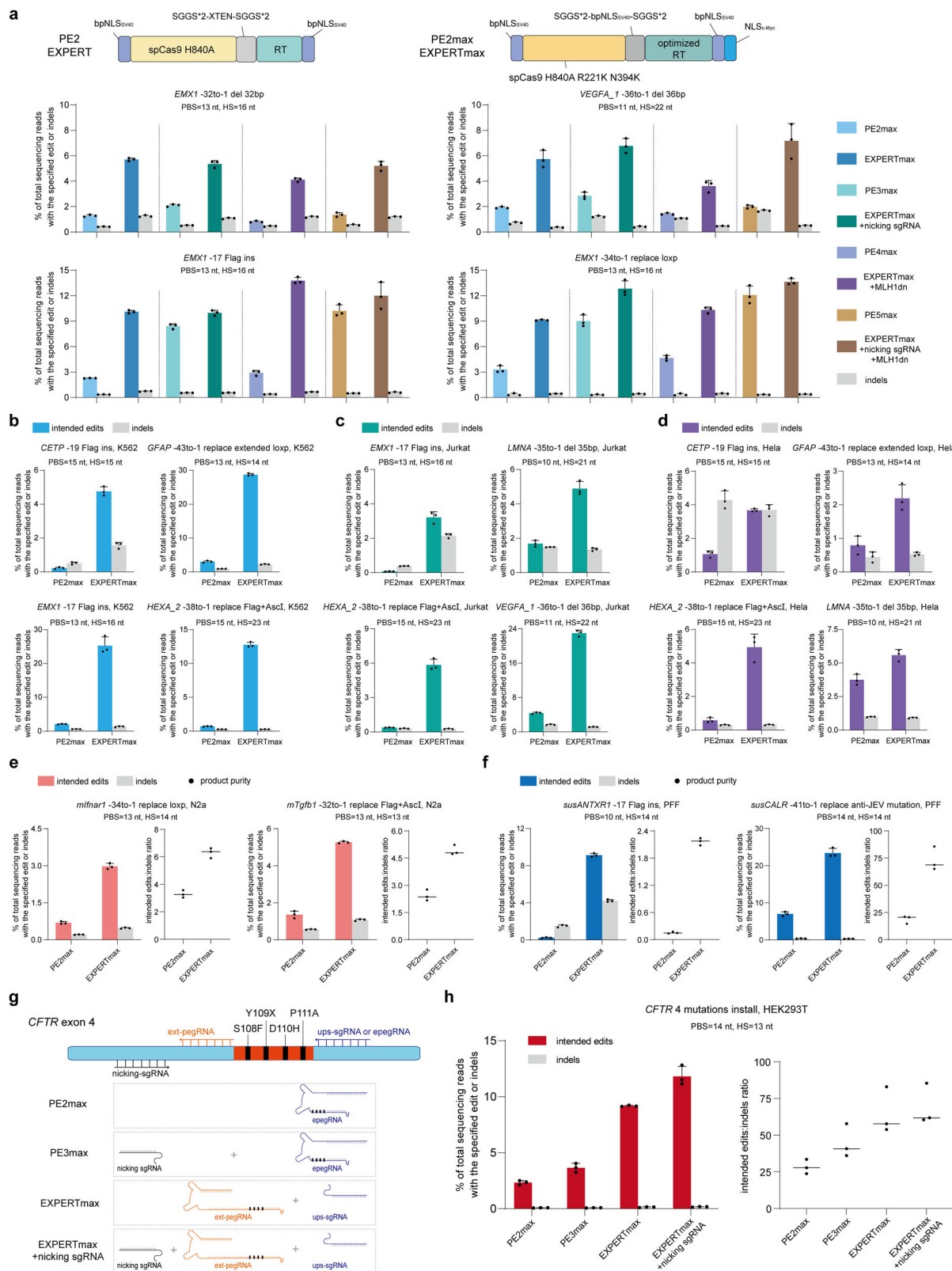

conceivable that the efficiency for even larger size fragment (i.e., >100 bp up to the kb range) may be achieved by introducing additional *cis* nicks to the system. This warrants future investigations.

We believe that the EXPERT strategy, which uses ups-sgRNA and ext-pegRNA, can be readily adopted by other PE systems, such as PE6 (PE6a-g)[33] and PE7[34] that have been recently reported, as long as they are based on nicking the target DNA. While we originally constructed

EXPERT based on PE2, we later constructed EXPERTmax by adding the ups-sgRNA and ext-pegRNA to PE2max. Such universal adaptability of the EXPERT strategy is a desirable feature given the continuous evolution of the PEs through other mechanisms such as pegRNA engineering, and protein engineering.

We want to point out some future research directions. There are universal challenges associated with the use of long pegRNA, including

**Fig. 5 | EXPERT strategy can enhance efficiency across various PE systems and can be applied to different cell types from multiple species. a** Frequencies of intended edits and indels introduced by PE2max, PE3max, PE4max, PE5max and their corresponding EXPERTmax systems. Additional mismatches were introduced in the insertion-type edits. Bars represent the mean of $n = 3$ independent biological replicates. Data are presented as mean ± s.d. **b** Frequencies of intended edits and indels introduced by PE2max and EXPERTmax in K562 cells. Additional mismatches were introduced in the insertion-type edits. Bars represent the mean of $n = 3$ independent biological replicates. Data are presented as mean ± s.d. **c** Frequencies of intended edits and indels introduced by PE2max and EXPERTmax in Jurkat cells. Additional mismatches were introduced in the insertion-type edits. Bars represent the mean of $n = 3$ independent biological replicates. Data are presented as mean ± s.d. **d** Frequencies of intended edits and indels introduced by PE2max and EXPERTmax in Hela cells. Additional mismatches were introduced in the insertion-type edits. Bars represent the mean of $n = 3$ independent biological replicates. Data are presented as mean ± s.d. **e** Frequencies of intended edits, indels, and product purity introduced by PE2max and EXPERTmax in N2a cells. Bars represent the mean of $n = 3$ independent biological replicates. Data are presented as mean ± s.d. **f** Frequencies of intended edits, indels, and product purity introduced by PE2max and EXPERTmax in PFF cells. Additional mismatches were introduced in the insertion-type edits. Bars represent the mean of $n = 3$ independent biological replicates. Data are presented as mean ± s.d. **g** Schematic diagram of complex mutations in *CFTR* exon 4. The intended edits that carried four mutations were performed in HEK293T cells using PE2max, PE3max, EXPERTmax, and EXPERTmax + nicking sgRNA, respectively. **h** Frequencies of intended edits, indels, and product purity introduced by PE2max, PE3max, EXPERTmax, and EXPERTmax + nicking sgRNA, respectively. Bars represent the mean of $n = 3$ independent biological replicates. Data are presented as mean ± s.d. All sequencing data were collected from transfection-positive cells. Source data are provided as a Source Data file.

susceptibility to exonuclease degradation, propensity to form secondary structures (especially for small edits), and difficulties in chemical synthesis. This is especially the case for EXPERT because compared to the pegRNA used in twinPE, the ext-pegRNA in EXPERT is ~15 bp longer. The propensity of longer pegRNA forming secondary structures may be alleviated through the introduction of mismatches which have been attempted and proven effective[11,24], whereas incorporating protective motifs into the pegRNA will help to address the susceptibility to exonuclease degradation problem as suggested by some studies[5-7], which should be tested to further improve the EXPERT. We also want to point out that EXPERT is suitable for large fragment edits (large sequence replacements, deletions, or insertions) and further improvements are needed for small edits. Until then, other tools such as base editors and PE3 are good choices for single-base substitutions and small edits. It also should be further noted that the EXPERT does not reduce the gRNA-independent OT effects, although it does not increase the OT effects either. How to minimize gRNA-independent OT effects is a general challenge for PE research which remains to be addressed.

In conclusion, we report the development of a PE tool EXPERT that holds tremendous potential for applications in life sciences.

## Methods

### Plasmids construction

The pegRNA-expressing cassette and puromycin-expressing sequence were amplified from pU6-pegRNA-GG-acceptor (Addgene: 132777) and pKLV2-U6gRNA5(BbsI)-PGKpuro2AZsG-W (Addgene: 67975), respectively. These components were then cloned into the pCMV-PE2 plasmid (Addgene: 132775) to generate the PE2-U6-puro vector. The Csy4 expression cassette was synthesized (GenScript) and then cloned into the PE2-U6-puro plasmid to generate the PE2-U6-puro-Csy4 vector. The PGK-Puro-T2A-ZsGreen sequence were amplified from pKLV2-U6gRNA5(BbsI)-PGKpuro2AZsG-W (Addgene: 67975) and then were cloned into pCMV-PEmax (Addgene: 174820) and pCMV-PEmax-P2A-MLH1dn (Addgene: 174828) to generate the pCMV-PEmax-puro-ZsGreen vector and the pCMV-PE4max-puro-ZsGreen vector, respectively. PCR was performed using the KAPA HiFi PCR Kit (Roche). Gel extraction of the amplified DNA fragments was performed using FastPure Gel DNA Extraction Mini Kit (Vazyme). The ClonExpress II One Step Cloning Kit (Vazyme) was used to construct PE2-U6-puro, PE2-U6-puro-Csy4, pCMV-PEmax-puro-ZsGreen, and pCMV-PE4max-puro-ZsGreen vectors.

Three versions of EXPERT vectors were utilized in this study. Schematic diagrams depicting the structure of these vectors are presented in Supplementary Fig. 19. Of note, both the ext-pegRNA and ups-sgRNA are expressed in the same one vector in all versions. The main difference in plasmid vectors between PE and EXPERT is that the EXPERT vector includes an additional guide RNA expression module. Overall, EXPERT v1 version was used in Figs. 1b and 2; EXPERT v3

version was used in Figs. 1c, 2g, 3, 4 and 5. All ext-pegRNA, ups-sgRNA, and pegRNA sequences were synthesized and cloned into PE2-U6-puro (digested with AarI), PE2-U6-puro-Csy4 (digested with AarI), pCMV-PEmax-puro-ZsGreen (digested with NruI), and pCMV-PE4max-puro-ZsGreen (digested with NruI) by Tsingke Biological Technology or GenScript to perform our experiments. The detailed information for the ext-pegRNA, ups-sgRNA, and pegRNA sequences used in this study is provided in Supplementary Data 1. The detailed sequences of all constructs are listed in Supplementary Data 2.

### Pre-screening of twinPE pegRNA pairs

A series of twinPE pegRNA pairs for these different loci were evaluated, and shown in Supplementary Fig. 11. Specifically, for each locus, we tested various combinations of pegRNA pairs, with the PBS of each pegRNA designed using the online tool pegFinder[35]. The detailed sequence information for these pegRNAs is provided in Supplementary Data 1. The best-performing twinPE pegRNA pairs, with a precise editing efficiency of at least 2% and the highest product purity, for each locus were selected and used for comparison with EXPERT.

### Cell culture, transfection, and genomic DNA isolation

HEK293T (ATCC, CRL-11268) and 293T-reporter cells were maintained in DMEM (Gibco) + 10% FBS (CELLiGENT) + 1% penicillin-streptomycin (Gibco); K562 (ATCC, CRL-3344) and Jurkat (ATCC, TIB-152) cells were maintained in RPMI Medium 1640 (Gibco) + 10% FBS (CELLiGENT) + 1% penicillin-streptomycin (Gibco); Hela (Procell, CL-0101), HFL1 (Procell, CL-0106) and N2a (Procell, CL-0168) cells were kindly provided by Procell Life Science & Technology Co., Ltd. Hela (Procell) and N2a (Procell) cells were maintained in MEM (Procell) + 10% FBS (CELLiGENT) + 1% penicillin-streptomycin (Gibco). HFL1 (Procell) cells were maintained in Ham's F-12K (Procell) + 10% FBS (CELLiGENT) + 1% penicillin-streptomycin (Gibco); PFF (isolated by our laboratory from a 35-day-old pig embryo) cells were maintained in DMEM (Gibco) + 10% FBS (Gibco) + 1% penicillin-streptomycin (Gibco). All these cells were cultured at 37 °C with 5% $CO_2$ and regularly tested to exclude mycoplasma contamination.

For transfection assays, HEK293T, Hela, and N2a cells were seeded into 6-well plates and transfected (~80% confluent) with 2.5 μg plasmid using JetPRIME (PolyPlus) according to the manufacturer's instructions. K562, Jurkat, HFL1, and PFF cells were transfected using an electroporation instrument (Celetrix). Briefly, a total of $1 \times 10^6$ cells were transfected with 2.5 μg plasmid using a 20 μL electroporation cuvette where the cells were shocked with 520 V.

All data in this study were collected through sequencing of transfection-positive cells. K562, Jurkat, and HFL1 cells were enriched with positive cells using flow cytometry sorting. Specifically, At 96 h post-transfection, GFP-positive cells (transfection-positive cells) were FACS-enriched (BD Aria™ III) for subsequent experiments. HEK293T, Hela, N2a, and PFF cells were enriched by adding puromycin

(InvivoGen) to the medium at the final concentration of 2 µg/mL, 24 h post-transfection. At 96 h post-transfection, transfection-positive cells were collected and lysed with freshly prepared lysis buffer and incubated at 65 °C for 40 min, followed by incubation at 95 °C for 15 min.

## Flow cytometry analysis or sorting

Flow cytometry analysis or sorting was performed 96 h post-transfection. 293T-reporter cells were collected after PBS washing and trypsin digestion and resuspended in PBS for flow cytometry analysis (Beckman CytoFLEX). Data were analyzed by CytExpert software. For Flow cytometry sorting, K562, Jurkat, and HFL1 cells were collected after PBS washing, resuspended in PBS, and sorted using a BD Aria™ III flow cytometer, followed by enrichment of GFP-positive cells.

## Deep sequencing and data analysis

The resulting cell lysis solution was subjected to PCR amplification of the target region using specific primers (Supplementary Data 3) with barcodes. Equal amounts of PCR products were pooled, purified, and commercially sequenced (Annoroad) using the NovaSeq platform. The obtained amplicon sequencing reads (FASTQ files) were then demultiplexed using bcl2fastq (Illumina) and analyzed by aligning the amplicon reads to a reference sequence by CRISPResso2[36]. To quantify the frequency of precise editing and indels, CRISPResso2 was run in the HDR mode. The precise editing efficiencies were calculated as the ratio of the number of desired reads to the total number of reads, while indel rates were calculated as the ratio of the number of indel-containing reads to the total number of reads.

## Analysis of genome-wide off-target editing

Genomic DNA was extracted from transfected and wild-type cells using the TIANapm Genomic DNA kit (TIANGEN) according to the manufacturer's instructions. WGS was performed by using the MGI DNBSEQ-T7 platform (Annoroad). Raw data was processed by SOAPnuke with default parameters to ensure the quality of data used in further analysis[37]. The obtained clean reads were mapped to the reference genome GRCH38.p12 from the National Center for Biotechnology Information (NCBI, https://www.ncbi.nlm.nih.gov/datasets/genome/GCF_000001405.38/) by Burrows-Wheeler aligner[38]. SAMtools were performed to sort reads and remove low-quality reads[39]. Duplicate reads from PCR were detected and removed by MarkDuplicates module of Genome Analysis Toolkit (GATK)[40]. The HaplotypeCaller module of GATK was performed for base substitutions and indels calling. The obtained variants were filtered by VariantFiltration module of GATK. The filtering settings were as follows: Base Substitution: $QD < 2.0$, $ReadPosRankSum < -8.0$, $FS > 60.0$, $QUAL < 30.0$, $DP < 4.0$, $MQ < 40.0$, $MappingQualityRankSum < -12.5$ and INDEL: $QD < 2.0$, $ReadPosRankSum < -20.0$, $FS > 200.0$, $QUAL < 30.0$, $DP < 4.0$. The number of genome-wide base substitutions and indels are listed in Supplementary Data 4. Potential gRNA-dependent off-target sites were predicted by Cas-OFFinder[41], allowing up to five mismatches. Off-target sites identified (Supplementary Data 4) by WGS shown in Supplementary Fig. 12 were randomly selected from genome-wide base substitutions.

## Statistics and reproducibility

At least three biological replicates were conducted per experiment to ensure the reliability of the data. Data are presented as mean ± s.d. Statistical analyses were performed with GraphPad Prism (version 9). To determine the statistical significance of a few results, the two-tailed Student's t-test was employed.

## Reporting summary

Further information on research design is available in the Nature Portfolio Reporting Summary linked to this article.

## Data availability

All sequencing data have been deposited in the NCBI Sequence Read Archive (SRA) under the BioProject Accession number PRJNA1203942. Source data are provided as a Source Data file. Source data are provided with this paper.

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

## Acknowledgements

We thank Prof. Xingxu Huang for his guidance on this work. We also appreciate the assistance of Yan Wang, Yuan Sun, and Yinqiu Wang from the Institute of Hydrobiology, Chinese Academy of Sciences in conducting the flow cytometry analysis or sorting. This work was supported by the National Key R&D Program of China (2021YFA0805903-3, J.R. and H.W.), the Foundation for Innovative Research Groups of the National Natural Science Foundation of China (32221005, S.Z., X.L., and S.X.), Major Program (JD) of Hubei Province (2023BAA029, S.Z., X.X. (Xuewen Xu), and J.R.), STI2030-Major Projects (2023ZD0404703, X.X. (Xuewen Xu), S.Z., and J.R.) and the earmarked fund for CARS (CARS-35, S.Z.).

## Author contributions

S.Z., J.R., and Y.X. designed the project. Most of the experimental work was conducted by Y.X., Y.S., and R.H., with minor contributions from X.H., M.L., X.X. (Xiaoning Xi), and Z.L. S.L., H.W., S.X., and X.X. (Xuewen Xu) analyzed the data. J.R., Y.X., J.X., J.Z., and K.L. prepared the figures and wrote the manuscript. X.L., J.R., and S.Z. supervised the project. The manuscript was reviewed and approved by all co-authors.

## Competing interests

S.Z., J.R., Y.X., X.L., Y.S., R.H., X.X. (Xiaoning Xi), X.H., and S.X. have filed two patent applications related to this work through Huazhong Agricultural University (2023115813808, 2023115813795). The remaining authors declare no competing interests.
