## [Transparent Peer Review file · Nature Communications]

EXPERT expands prime editing efficiency and range of large fragment edits

Corresponding Author: Dr Jinxue Ruan

Version 0:

Reviewer comments:

Reviewer #1

(Remarks to the Author)

Xiong et al reported the development of EXPERT system for increasing the prime editing scope. By introducing a nicking guide upstream of the pegRNA nick site and a PBS that binds to the nicked DNA single strand upstream, authors hoped that EXPERT can expand the intended PE edits upstream, where the canonical PE system cannot reach. However, though the upstream nicking design is new, but in the manuscript, authors didn't show EXPERT can translate the edit types (all transition and transversion point mutations, small insertions and deletions, and combinations of the base substitution and indels) that canonical PE can efficiently perform to the upstream region. Deletion, insertion, and replacement edits have been shown with EXPERT, but the published methods, for example: TwinPE, Prime-Del, GRAND etc., can already execute with these types of edits and are not limited by the nicking direction because of the usage of a pair of pegRNAs. Besides, there are a few problems and concerns that authors need to address to demonstrate the potentials of EXPERT as a useful tool to the genome editing field and a broad research community.

1. The edits in the article are mainly sequence replacement and deletion. Authors lack the data showing EXPERT's performance on small indels and insertions and base substitutions, and thus limit the versatility of EXPERT. For sequence replacement and deletion, paired pegRNAs technologies nowadays can efficiently execute with the choices of suitable PAMs upstream and downstream of the target DNA site. Additionally, when performing small types of edits, there will be higher level of homology between the RT (reverse transcriptase) template and ext-pegRNA's spacer (complementarity between up-sgRNA and up-sgRNA), which could lead to secondary structure within the ext-pegRNA and poison the assembly of prime editor and ext-pegRNA complex. That concern has been raised in the recent publications (Nelson et al NBT 2021 and Ponnieselvan et al NAR 2023).
2. How does the homology sequence in the 3' flap (encoded in the RT template) compete with the 5' flap during Flap Equilibration step and how does the length of homology sequence affect the editing percentage? It is important to clarify it in EXPERT design since this is related to the overall architecture of ext-pegRNA.
3. What is the editing efficiency (% of mCherry+ cells) if only the premature stop codon gets corrected without introducing synonymous AA sequence?
4. In figure 2e and f, authors showed that introducing mismatches can improve EXPERT editing efficiency. One issue is that the improvement and the best performance of EXPERT usually requires many mismatches in a very limited editing window (10 base pairs needed for VEGFA -34 -17 del edit; 11 for HEK4_1 -32 to -14 replacement; 7 for HEK4 -18 to -1). One advantage for classic PE system (PE2-PE6) is to install the edit and target disease mutations precisely. Here, when target gene coding sequence with EXPERT, apparently users need to install quite a high % (14 bp-window requires at least 7 mismatches as shown in the figure, >50%) of silent mutations to reach its optimal performance. In general, considering the codon degeneracy, it is very hard to achieve that high level of codon recoding. Therefore, it is very limited, comparing to PE4 strategy, which introduces a smaller number of silent mutations to evade mismatch repair pathway and achieves maximum editing benefits.
5. In figure 2 and other figures, there is no information about the primer binding sequence (PBS) of ext-pegRNA. What is the length of PBS used in this figure and others? How does PBS length impact the editing percentage? Since PBS plays a very important role in initiating reverse transcription and ensuring efficient prime editing efficiency, please demonstrate how it can impact EXPERT performance.
6. When perform large insertion edit with EXPERT, it requires a very long ext-pegRNA, which is more prone to exonuclease trimming and secondary structure forming. In twinPE and similar methods, a partial insertion sequence can be encoded in each of the dual pegRNAs and then achieve the full-size, large insertion at the target site via partial sequence

complementarity between the two newly synthesized 3' flaps. The long ext-pegRNA requirement can also become a problem when editing primary cells that favor synthetic pegRNA and RNA components due to the DNA plasmid toxicity because chemical synthesis of long pegRNAs is very challenging to the vendors.

7. In figure 4 and 5, the comparison of EXPAND to PE2 needs to be carefully devised to avoid unfairness. In figure 4a authors uses ups-sgRNA target site directly as pegRNA target sequence. If that is the case, authors should also perform the comparison using the optimal pegRNA from PE2 and PE3 edits shown in Anzalone et al., Nature 2019 (Figure 2-5) and find a downstream DNA target for ext-pegRNA. This can help to determine whether ext-pegRNA's improvement on editing % can be generally observed. Alternatively, if authors must pick the sites that is more frequently used for EXPERT, at least you should screen a few upstream protospacer sequence, optimize PBS and RTT for canonical PE, which is the recommended procedure in the Anzalone et al., Nature 2019 and other recently published papers on experimental and computational design of pegRNA.

Reviewer #2

(Remarks to the Author)

Comments to the Authors

The manuscript reports on modifications of prime editing technology that the authors find can improve editing performance in their studies. The modifications were rigorously tested using a variety of cell lines and targets. Overall the manuscript was clear and easy to interpret and understand.

Major comments:

The use of the word "replication" could use further clarification. Are the replicates shown from exactly the same experiment (same cells, same transfection, same day, same harvest times), or are they from completely independent experiments conducted on three separate days. In other words, are they technical or biological replicates? In many cases the error bars are extremely small, and so this is important to clarify.

Lines 113-115. It seems as if this concept would be better placed after an introduction to the technology (after Figure 1 discussion?). There are a lot of studies described in Supplementary Figure 1. Perhaps they deserved their own paragraph of discussion to walk the reader through these studies.

Minor:

The editors may not allow for reference to figures in the introduction.

Abbreviations might need to be defined better upon first use.

Line 72. It might be helpful to provide the reported editing efficiencies for the available Pes described, just to provide context for the current work.

Line 91. "than that by" should be "compared to".

Figure 1. The editable portion is in "green" in the canonical PE side of Figure 1a. Should you make the upstream portion green as well in the EXPERT side. This would help to highlight that the editable portion of the target sequence has expanded (or at least, don't keep both as red). Can you define the blue and purple and light blue structures in this figure?

Figure 1b legend is not clear enough (what is being represented "respectively?"). Are you saying the canonical PE cannot edit, but the EXPERT will edit at two sites? What is RTT in this figure?

It is not clear why VEGFA was used in Figure 1. Was there some type of rationale to utilize this target sequence from the beginning? It is not critical to know, but would help to explain the experiment conducted in context.

It never became completely clear if two guide RNAs are expressed from the vectors. Could you expand a bit on exactly what the plasmid vectors look like in PE vs EXPERT-PE?

Line 674. "that using" should be "that used".

Line 682. As the authors have done for "Figure 2c" it would be helpful to the reader if the consequences of the hybridization of the 5' flap are explained.

Lines 163-168. The text is a bit confusing as it is not really obvious from the figure that that intended site is "away" from the nick site. Was this site chosen because the frequency was very low, thus you could then see a signal when you added the mismatch?

Line 259-263. While there was no difference between the experimental groups, each showed some gRNA-independent off target effects? How many were seen? Is this still a problem for the technology? It seems as if the EXPERT system did not reduce the number of off target effects.

Figure 4 legend title does not reflect the various aspects of the figure, especially the off-target effects. Should there be a broader title chosen?

Reviewer #3

(Remarks to the Author)

Brief intro to paper

This is potentially an interesting and novel adaptation of Prime Editing called EXPERT. Briefly, it builds on canonical PE (also known as PE2) by adding an extra guide RNA and an "extra" elongated 3' end of the pegRNA (I say "extra" elongated pegRNA as the pegRNA is itself already an elongation of the 3' end of a standard CRISPR guide RNA).

The consequence of positioning the gRNA upstream of the nick site created by PE2/pegRNA is that an additional 3' flap is created which provides an alternate primer binding site (PBS). And since the "extra" elongated pegRNA has a PBS to bind this additional 3' flap, that's where the editing starts.

Not to diminish the novelty and work presented herein, this isn't a whole new system, rather it's still PE2 but with a nicking

gRNA in a different position to the PE3 system, and an “extra” elongated pegRNA. This does allow the creation of longer edits than previously reported, and that in itself is a very important development. As authors point out, the use of the gRNA to nick on the same strand (a cis site), is important in reducing the likelihood of indels. In many cases, this appears to work very well, but as mentioned below, there are few examples, where I’m not convinced (yet) that this is always the case. But it’s a big improvement.

My specific comments are as follows.

Abstract

Ln 36 – insert the word “region” after “targeting the”

Ln 44 – authors should include a statement explaining their idea why cis nicking should reduce indels

Intro (minor comments unless otherwise stated)

Ln 49 – insert “*Strep. pyogenes*” before “Cas9” (they mention source of RT on Ln 50 so should include source of Cas9).

Ln51-55 – may be better as two sentences. First part is wrong; as authors know, the pegRNA/Cas9-RT creates a nick which liberates a 3’ flap which binds the PBS on the 3’ end of the pegRNA. They should also state the template part of the pegRNA is known as the RTT or edit sequence.

Ln 60 – I don’t think it expands the scope, rather it specifically extends the length of edits that can be done by PE. To justify this request, PE extended the scope of editing relative to BE; BE could only do C>T or A>G, whereas PE could do all 12 SNP changes plus inserts and deletions. Thus, EXPERT doesn’t increase the scope, just the length.

Ln 62 – TJ-PE was INSPIRED by the activity of endogenous retrotransposons, but to best of my knowledge uses the Cas9/RT prime editor – it does not “harness the activity of endogenous retrotransposons”

Ln67-68 – edit slightly, as written it’s literally incorrect. E.g. insert “alter” or “change” before “certain amino residues”

Ln71-75 – this is an important point to be considered. EXPERT is an interesting system, but I would invite the authors to rephrase this slightly; I’ll phrase my point as carefully as I can. I don’t think EXPERT can edit upstream, rather it’s shifting the start site upstream so the overall effect is a longer editing tract. I like this strategy, but I think it’s slightly incorrect as written.

LN78-80 – just a comment, this is a good point well made, and potentially an important feature of EXPERT relative to most (all?) other PE systems to date.

LN 91-92 – whilst they have a specific example of an increase of 122-fold relative to PE2, I don’t think that’s a completely fair representation as many times, the increase is no where near this. Maybe, stating an average and a range, e.g. SEM or 95% confidence level etc. would better portray the increase in activity. Additionally, it may be fairer to compare the fold increase with the PE3 values which also uses a second gRNA, although I would concede that PE3 uses the nicking guide to boost efficiency in a potentially different way.

Ln 92-93 – again, to be strictly correct, this isn’t really changing the function of PE, and it’s not really changing the versatility, but it does increase the length of sequences that can be corrected.

Results

Ln106/Figure 1b – the color scheme in 1b doesn’t match the color scheme in 1a – i.e. 1a shows upstream edits in red, but 1b shows the first edit in green, but it’s a upstream edit. Then the next upstream edit is in blue! Maybe both should be in red, one full color, the other with a striped pattern maybe?

Also, would a red triangle to shown the nick site help – most papers use that symbol?

Ln 109 – “In the work” doesn’t make sense. Do they mean, “In this study...” or “In our strategy ...”?

Ln118-125 and Figure 1 – not clear from the figure or the text if the two edits are performed by two different pegRNAs, or if one peg can perform both edits – can this be clarified please.

Ln126 – the prime editing field suffers from a challenge of nomenclature with PE being used to refer to both the process (prime editing) and the editors (prime editors) – thus I struggle with the first sentence. There is no difference between the prime editor used in canonical prime editing, and expert – the difference is one uses only a pegRNA whereas the other uses an “extra’ elongated pegRNA and a nicking gRNA.

Simplest solution could be to replace “canonical PE” with “canonical prime editing strategies”?

Ln 135 – insufficient data (yet) to support “proves” so use “ supports the hypothesis” instead.

Ln 147 – I agree the highest is achieved at 38-51 nt, but it’s only 2-4% so this should also be mentioned in the text as well as the figure. Authors claim earlier a 122-fold improvement, but in many cases it’s much lower, so I think referring to the % activity in the text as they go a long is important, and will also help in finding a figure that better reflects the overall fold

improvement of EXPERT.

Ln 148-152 – as another general comment, they need to comment on indels – they are clearly present, in some cases at quite high levels, but in the abstract, intro and discussion, indels are described as very low – the data in the text doesn't always support these “blanket” statements about the low levels of indels in other parts of the manuscript.

Can they also comment why the editing with 24nt spacing is so low – is it simply a steric hindrance issue?

Ln 156-162 – this concept is nicely introduced – it's an intriguing concept.

Ln 170-183 – the VEGF and HEK4 editing profiles and indel profiles are quite different – any idea why, and they don't really comment upon this – they should. For example, VEGF has almost no indels, but HEK4 has lots. Also, why do they show a “purity profile” in Fig2h, but not Fig 2e?

Also, why such a high level of indels in Fig 2h, and why does it increase with the number of mismatches.

And, why did they do continuous tracts of mismatches – did they test replacing, for example, every second base?

Ln208-209. Suggest minor re-write to emphasize that “helper gRNA” is a new term they are proposing.

Was “To improve EXPERT's capacity to insert or replace long fragment, we included a Helper gRNA positioned between the ups-sgRNA and ext-pegRNA (Fig. 3c)”.

Suggest “To improve EXPERT's capacity to insert or replace long fragment, we included an extra gRNA, positioned between the ups-sgRNA and ext-pegRNA, which we refer to as the Helper gRNA(Fig. 3c)”.

Ln 210-213 – suggest minor reordering of sentence to better reflect the most likely real life order of events

Was, “...which would facilitate the detachment of the original DNA strand, and may recruit an additional nCas9-RT enzyme between the original two nicks thereby increasing the overall reverse transcription”.

Suggest, “... which may recruit an additional nCas9-RT enzyme between the original two nicks, and then facilitate the detachment of the original DNA strand, and thereby increasing the overall reverse transcription”.

Ln 220-223 the statement about indels is too strong regarding their prediction that nicking in cis “confirms” their speculation. Indeed, they have quite a few indels in Fig 2 and 3b. A nuanced version of this statement is suggested.

Ln 235 – it might be useful to discuss why they chose to compare EXPERT with PE2 rather than PE3, since both PE3 and EXPERT use a pegRNA and nicking guide – it might have been a fairer comparison when looking at absolute levels of editing, and also indel formation.

Figure 4a – this is a really nice set of data, but two comments. First, given this is the most comprehensive comparison of PE2 and EXPERT, I don't see how this data leads them to the 122-fold increase in editing relative to PE2 claim in the abstract?

Second, and this is a minor point, but could help in interpretation of the data. They use three different colors for inserts, deletions and replacements, but in the graphs to analyze (4b and 4c), the colors are removed – it would be interesting to see if they inserts, indels and replacements are grouped or randomly distributed – simply asking for data points in Fig 4b/c to have the same colors.

Ln 272-282 – suggest supplementary figure to show purity ratios for EMX data

Ln283-284 – I agree, this data does validate the claim that this approach should work with most other prime editors currently available and others that are generated in future. Would be cool to see in the future how EXPERT works with some of the new editors such as PE6, in particular the PE6 version that works best with installing longer edits? Authors could speculate on this or similar ideas in discussion?

Ln 313-314 – given my comments on indels earlier, I think this statement should be modified slightly.

Ln 338 – Figure 5f – suggest minor adjustment to figure 5f such that the epegRNA is in the same place in all four lines of the figure – also, the approximate position of the nicking site for the epegRNA, the ups-gRNA and nicking guide should be shown on exon 4 of the CFTR diagram.

Ln345-346 – that statement is too strong for the limited amount of genetic disease data, it should be modified.

Ln350-352 – does the phrase “the 3' extension binds to the trans strand of the PBS site” make sense – I think it needs modification.

Ln 355 – “region” is better than “sides”

Ln357 – a 122-fold increase is claimed here and in the abstract, but I don't see a justification for it – can it be provided/clarified or removed.

Ln 358 – should that sentence end with “relative to PE2”?

Ln 363 – I don't think 2.9% indels is extremely low – also, 2.9% editing seems to conflict with statement on ln357/358 about low indels?

Ln 391-393 good point.

Version 1:

Reviewer comments:

Reviewer #1

(Remarks to the Author)

Xiong et al have provided additional data to strengthen this manuscript. There are a few issues that still remain and need to be addressed.

- 1) The authors did not show EXPERT's performance on single-base substitutions in response 1.1.1.
- 2) The number of twinPE pegRNA pairs screened was not specified. Dual pegRNA methods, including twinPE, require screening and optimization for high efficiency and purity at the genomic locus. Please include this information for a fair comparison between twinPE and EXPERT, which also requires optimization.
- 3) In R1.7, the edits primarily involve large sequence replacements, deletions, or insertions. Since prime editing (as per Anzalone et al., Nature, 2019) is highly efficient at single-base substitutions and small indels, the authors should include single-base substitution edits using PE3 (1 pegRNA + 1 nicking sgRNA) and compare with EXPERT (1 pegRNA + 1 nicking sgRNA).
- 4) The revised manuscript lacks sufficient evidence to support the claim that "EXPERT enhances editing efficiency of all types." Specifically, single-base substitution comparisons were made between PE2 (using one pegRNA) and EXPERT, which involved additional optimization. PE3 and PE5, with similar optimization, should be used to substantiate this claim. Authors should revise their statement to accurately describe the advancement of EXPERT to the field.
- 5) Cellular repair machinery varies among cell types (e.g., transformed vs. primary cells). Data from non-transformed or therapeutically relevant cells (e.g., T cells, iPSCs, primary fibroblasts) are needed to evaluate EXPERT's broader potential in life sciences.

Reviewer #2

(Remarks to the Author)

Thank you for carefully considering the suggestions. The responses were complete.

Reviewer #3

(Remarks to the Author)

All questions addressed thoroughly, no further comments

Version 2:

Reviewer comments:

Reviewer #1

(Remarks to the Author)

Thank you for conducting additional experiments to address my comments and strengthen the manuscript. I suggest that the authors exercise caution when drawing conclusions from their data. While the manuscript demonstrates clear strengths—such as the ability of EXPERT to target the upstream region of the ext-pegRNA, offering opportunities to improve efficiency and reduce indels for certain edits. Some conclusions in the rebuttal and manuscript text are not stated properly. I recommend that the authors revise the manuscript main text accordingly. Below, I provide three examples for improvement:

In R1.3 (Supplementary Figure 5b):

You stated, "Interestingly, however, the unintended on-target indel rates were lower with EXPERT than with PE3." Readers may not fully appreciate this conclusion given that the on-target editing efficiency for EXPERT is very low, 4- to 12-fold less efficient than PE3. Moreover, the fold improvement in indel reduction with EXPERT is only 2- to 3-fold better than PE3. If you wish to discuss product purity, I recommend using the editing specificity ratio, which is commonly reported in genome editing literature.

Regarding the twin prime editing pegRNA screening:

In Anzalone et al. NBT 2022, the authors screened multiple pairs of pegRNAs, PBS lengths, and various combinations of pegRNAs. Liu Lab conducted a more comprehensive evaluation of pegRNAs to select the best twin prime pegRNA pairs that demonstrate low indels and high efficiency for downstream applications. I understand that performing pegRNA screening at the same scale and making thorough comparisons is challenging in this manuscript. However, it would be more informative to acknowledge the specific parameters evaluated in your twin prime editing pegRNA screen and state a more comprehensive comparison would be recommended for further investigation.

Additional cell line data and method clarification:

The additional data from transformed cell lines and primary cell HFL1 experiments support the applicability of EXPERT in other cell types. However, the figure legend should specify whether the % editing shown in the bar graph represents FACS-enriched, transfected cells. If so, what is the non-selected, bulk cell editing efficiency? Please update the corresponding Methods section, as the current description of the FACS selection protocol is insufficient for replication.

I recommend publication of this manuscript once the authors address the comments outlined above and carefully revise manuscript main text.

Point-by-point response to reviewers' comments

Reviewer #1 (R1):

R1 general comments: Xiong et al reported the development of EXPERT system for increasing the prime editing scope. By introducing a nicking guide upstream of the pegRNA nick site and a PBS that binds to the nicked DNA single strand upstream, authors hoped that EXPERT can expand the intended PE edits upstream, where the canonical PE system cannot reach.

However, though the upstream nicking design is new, but in the manuscript, authors didn't show EXPERT can translate the edit types (all transition and transversion point mutations, small insertions and deletions, and combinations of the base substitution and indels) that canonical PE can(not) efficiently perform to the upstream region. Deletion, insertion, and replacement edits have been shown with EXPERT, but the published methods, for example: TwinPE, Prime-Del, GRAND etc., can already execute with these types of edits and are not limited by the nicking direction because of the usage of a pair of pegRNAs. Besides, there are a few problems and concerns that authors need to address to demonstrate the potentials of EXPERT as a useful tool to the genome editing field and a broad research community.

Response: Thank for your valuable feedback. Please see our responses to your specific comments below.

R1.1: The edits in the article are mainly sequence replacement and deletion. Authors lack the data showing EXPERT's performance on small indels and insertions and base substitutions, and thus limit the versatility of EXPERT. For sequence replacement and deletion, paired pegRNAs technologies nowadays can efficiently execute with the choices of suitable PAMs upstream and downstream of the target DNA site. Additionally, when performing small types of edits, there will be higher level of homology between the RT (reverse transcriptase) template and ext-pegRNA's spacer (complementarity between up-sgRNA and up-sgRNA), which could lead to secondary structure within the ext-pegRNA and poison the assembly of prime editor and ext-pegRNA complex. That concern has been raised in the recent publications (Nelson et al NBT 2021 and Ponnienselvan et al NAR 2023).

Response: Thank you for your professional comments.

R1.1.1 For comments about if EXPERT can achieve different types of small edits, we now conducted various types of small fragment editing experiments (including small indels and base substitutions), and demonstrated that EXPERT is capable of performing all these types of small edits in the upstream region of the ext-pegRNA nick (Supplementary Fig. 5). We have included the results in the revision.

Supplementary Fig. 5 Frequencies of intended small edits and indels were performed by EXPERT and EXPERT with introducing 3 mismatches. Bars represent the mean of $n = 3$ independent replicates.

R1.1.2 Regarding your comments that “for sequence replacement and deletion, paired pegRNAs technologies nowadays can efficiently execute with the choices of suitable PAMs upstream and downstream of the target DNA site”, we conducted additional experiments to directly compare the editing outcomes by EXPERT and the representative two-pegRNAs system twinPE. The results are summarized below and included in the revision.

For sequence insertion, deletion and replacement, twinPE is efficient. However, twinPE induces nicks on both DNA strands (referred to as trans nicks) and increases the likelihood of DNA double-strand breaks and subsequent indels. In contrast, EXPERT induces nicks on only one DNA strand (referred to as cis nicks). Here we conducted comparative experiments between EXPERT and twinPE across four loci. Our results show varied but comparable editing efficiencies between EXPERT and twinPE depending on the locus: EXPERT showed higher efficiency at the *VEGFA_1* locus, twinPE exhibited superiority at the *LMNA* and *HEK4* loci, and they performed similarly at the *VEGFA* locus (Fig. 4d). Nevertheless, in all loci tested, EXPERT induced indel rates are lower than those induced by twinPE. As a result, the purity achieved by EXPERT is consistently better than that by twinPE (Fig. 4d), which supports our hypothesis and demonstrates the potential advantage of EXPERT in gene editing applications.

Fig. 4d Frequencies of intended editing and indels introduced by twinPE and EXPERT at different sites. Bars represent the mean of n = 3 independent replicates.

R1.1.3 For comments about concerns over the potential secondary structure due to the high homology between the RT (reverse transcriptase) template and ext-pegRNA’s spacer in small edits, we agree that this should be considered. Indeed, our work reports that to enhance the editing efficiency of EXPERT for small edits, it is crucial to introduce additional mismatches to reduce the hybridization. Additionally, optimizing the base composition of pegRNA is also effective, as demonstrated by recent studies (Nelson et al NBT 2022 and Ponninselvan et al NAR 2023). We have added relevant discussion to this point in the Discussion.

Specifically we have revised the following:

Lines 262-269: “Lastly, we evaluated EXPERT in introducing small indels and base substitutions in the upstream region of the ext-pegRNA nick, with or without inclusion of mismatches (Supplementary Fig. 5). The results showed that EXPERT can perform all these types of small edits in the upstream region of the ext-pegRNA nick. Again, introducing mismatches significantly improved the editing efficiency, in line with our speculation that introduced mismatches reduces the homology between the ext-pegRNA and the genomic DNA 3’ Flap, thereby preventing their hybridization (Fig. 2c).” has been added.

Lines 348-368: “Comparison of EXPERT with twinPE

We next compared EXPERT with one two-pegRNAs system twinPE. EXPERT induces both nicks on the same DNA strand (i.e., *cis* nicks), whereas the twinPE induce one nick on each strands (referred to as *trans* nicks). We speculate that *trans* nicks, in comparison to *cis* nicks, increase the likelihood of DNA double-strand breaks and subsequent indels. To validate this speculation, we conducted comparative experiments between EXPERT and twinPE in four different edits at four different loci. Briefly they are: (i) *VEGFA* 36bp deletion and 78bp insertion, (ii) *LMNA* 36bp

replacement, (iii) *VEGFA_1* 37bp deletion and 103bp insertion, (iv) *HEK4* 37bp deletion and 79bp insertion. Detailed information of specific edits is shown in Fig. 4d.

Our findings show varied but comparable editing efficiencies between these two systems: EXPERT had higher efficiency at the *VEGFA_1* locus, whereas twinPE exhibited superiority at the *LMNA* and *HEK4* loci, and both systems performed similarly at the *VEGFA* locus (Fig. 4d). Importantly, purity analysis demonstrated significantly higher purity is achieved with EXPERT in all edits, ranging from 1.5 to 5.8-fold higher, than that by twinPE (Fig. 4d).

We also want to point that unlike the two-pegRNAs systems which require two separate PAM sequences (NGG) on both DNA strands, the EXPERT system only needs PAM sequences on one strand, which allows EXPERT to target additional regions. Such comparison experiment cannot be conducted because these regions are only targetable by EXPERT but not by twinPE or any two-pegRNAs systems.” had been added.

R1.2: How does the homology sequence in the 3' flap (encoded in the RT template) compete with the 5' flap during Flap Equilibration step and how does the length of homology sequence affect the editing percentage? It is important to clarify it in EXPERT design since this is related to the overall architecture of ext-pegRNA.

Response: Thank you for your feedback.

R1.2.1 With regard to the question about how does the homology sequence in the 3' Flap compete with the 5' Flap, as shown in Supplementary Fig. 2c, it's speculated that the 5' Flap of the genomic DNA would be excised by endogenous nucleases such as FEN1 or EXO1 (Anzalone et al Nature 2019). In EXPERT, the presence of two *cis* nicks may enhance the detachment of the original single-stranded DNA fragment from the genome, thereby promoting subsequent processes. Next, the homology sequence (We refer to this as HS, supplementary Figure 1) in the 3' Flap would complement to the genomic DNA, thereby driving the incorporation of the edited DNA strand (Anzalone et al Nature 2019). We have included a discussion in the revision.

R1.2.2 With regard to how does the length of homology sequence affect the editing rates, our data suggest that effective editing can be achieved with HS lengths ranging from 12 to 22 nt. When the HS length is shorter than 16 nt, there is only one locus with an editing efficiency greater than 6% (1 out of 12 or 8%). In contrast, when the HS length is 16 nt or greater, there are six loci with an editing efficiency exceeding 6% (6 out of 7 or 86%) (Supplementary Fig. 8). Therefore, we recommend an HS length of 16 nt or greater as the starting design. Following your suggestion, we now labeled the lengths of HS for each ext-pegRNA in the figures.

Supplementary Fig. 1 The structural components of ext-pegRNA.

Supplementary Fig. 8 The investigation of the optimal homology sequence (HS) lengths for EXPERT. When the HS length is shorter than 16 nt, there is only one locus with an editing efficiency greater than 6% (1 out of 12). In contrast, when the HS length is 16 nt or greater, there are six loci with an editing efficiency exceeding 6% (6 out of 7).

Specifically we have revised the following:

Lines 113-115: “The ext-pegRNA comprises a primer binding site (PBS) and a reverse transcriptase template (RTT). The RTT consists primarily of an edit sequence (ES) and a homologous sequence (HS) (Supplementary Fig. 1).” has been added.

Lines 128-130: “Mechanistically, we speculate that this is because EXPERT generates two *cis* nicks that enhances the detachment of the original single-stranded DNA fragment from the genome, thereby promoting subsequent processes (Supplementary Fig. 2c).” has been added.

Lines 283-290: “Subsequently, we conducted an investigation to determine the optimal length of HS. Analyses conducted on the editing efficiency of different HS lengths indicated that effective editing could be achieved with HS lengths ranging from 12 to 22 nt (Supplementary Fig. 8). When the HS length is shorter than 16 nt, there is only one locus with an editing efficiency greater than 6% (1 out of 12 or 8%). In contrast, when the HS length is 16 nt or greater, there are six loci with an editing efficiency exceeding 6% (6 out of 7 or 86%) (Supplementary Fig. 8). Therefore, we recommend an HS length of 16 nt or greater as the starting point in EXPERT design.” has been added.

R1.3: What is the editing efficiency (% of mCherry+ cells) if only the premature stop codon gets corrected without introducing synonymous AA sequence?

Response: Following your suggestion, we implemented a small edit strategy by correcting only the premature stop codon without introducing synonymous mutations. The editing efficiency by EXPERT is 5.6% (see below).

Editing efficiency with correction of premature stop codon only, without introducing mismatches. Bars represent the mean of n = 3 independent replicates.

R1.4: In figure 2e and f, authors showed that introducing mismatches can improve EXPERT editing efficiency. One issue is that the improvement and the best performance of EXPERT usually requires many mismatches in a very limited editing window (10 base pairs needed for VEGFA -34 -17 del edit; 11 for HEK4_1 -32 to -14 replacement; 7 for HEK4 -18 to -1). One advantage for classic PE system (PE2-PE6) is to install the edit and target disease mutations precisely. Here, when target gene coding sequence with EXPERT, apparently users need to install quite a high % (14 bp-window requires at least 7 mismatches as shown in the figure, >50%) of silent mutations to reach its optimal performance. In general, considering the codon degeneracy, it is very hard to achieve that high level of codon recoding. Therefore, it is very limited, comparing to PE4 strategy, which introduces a smaller number of silent mutations to evade mismatch repair pathway and achieves maximum editing benefits.

Response: Thank you for your insightful comment. In our previous experiments, we primarily utilized the design of consecutive mismatches, which sometimes has very high percentage of the mismatch rate as you pointed out.

In this revision, we tested another strategy to introduce mismatches, i.e., a mismatch every n nucleotides, where n = 3 or 5. Interestingly, our findings indicate that the editing efficiency of ext-pegRNA with mismatches introduced every 3 or 5 nucleotides closely resembles that of achieved by ext-pegRNA with long consecutive mismatches (Supplementary Fig. 7). By using the “1 in 5 mismatch”, the mismatch rate is decreased to 20%. We have included the outcomes of these interval mismatch experiments in the Results section of our manuscript.

Supplementary Fig. 7 Frequencies of intended editing and indels were performed by EXPERT at the indicated target sites under the different number of mismatches on ext-pegRNA region. Bars represent the mean of n = 3 independent replicates.

Specifically we have revised the following:

Lines 251-256: “We also tested another strategy to introduce mismatches. Specifically, the mismatches are introduced at intervals of every 3 and 5 nucleotides, respectively (Supplementary Fig. 7). The results indicate that the editing efficiency of ext-pegRNA with such mismatches closely resembles that observed with ext-pegRNA featuring 11 consecutive mismatches. These results suggest an alternative mismatch design of the ext-pegRNA for achieving effective editing with EXPERT.” has been added.

R1.5: In figure 2 and other figures, there is no information about the primer binding sequence (PBS) of ext-pegRNA. What is the length of PBS used in this figure and others? How does PBS length impact the editing percentage? Since PBS plays a very important role in initiating reverse transcription and ensuring efficient prime editing efficiency, please demonstrate how it can impact EXPERT performance.

Response: Thank you for your comments. We agree that PBS plays a crucial role in initiating reverse transcription and ensuring efficient prime editing. Our understanding is that you want to know how the “length of PBS” impacts EXPERT performance.

To answer this, we conducted additional experiments to investigate the impact of the PBS length on the EXPERT editing efficiencies at two loci: *VEGFA_1* and *PRNP*, and the results suggest that PBS lengths ranging from 9 to 16 nt all lead to efficient editing outcomes. From a practical point, we would recommend using PBS length of 12 nt, and avoid using PBS shorter than 9 nt. We have also updated all figure captions to include PBS length information as per your suggestion. The Results section now incorporates our experimental results on the impact of optimal PBS length.

Fig. 2g The impact of PBS length on editing efficiency at different sites. Bars represent the mean of n = 3 independent replicates.

Specifically we have revised the following:

Lines 272-282: “Optimization of PBS length and HS Length in EXPERT

PBS plays an important role in initiating reverse transcription and ensuring efficient prime editing efficiency^{1, 28}. To assess if the length of PBS affects the editing efficiency of EXPERT, experiments were conducted at two loci: the *VEGFA_1* and *PRNP* loci, with PBS length incremented by 1 nt ranging from 8 to 17 nt (Fig. 2g). The results indicate that PBS lengths ranging from 9 to 16 nt could lead to efficient editing outcomes: 11.64% to 15.41% at the *VEGFA_1* locus, and 2.42% to 5.53% at the *PRNP* locus. From a practical point, we would recommend using PBS length of 12 nt because at both loci this parameter led to the highest editing efficiency, and avoid using PBS shorter than 9 nt as apparently this drastically reduced the editing rate in one tested locus.” has been added.

R1.6: When perform large insertion edit with EXPERT, it requires a very long ext-pegRNA, which is more prone to exonuclease trimming and secondary structure forming. In twinPE and similar methods, a partial insertion sequence can be encoded in each of the dual pegRNAs and then achieve the full-size, large insertion at the target site via partial sequence complementarity between the two newly synthesized 3’ flaps. The long ext-pegRNA requirement can also become a problem when editing primary cells that favor synthetic pegRNA and RNA components due to the DNA plasmid toxicity because chemical synthesis of long pegRNAs is very challenging to the vendors.

Response: Thank you. In response to these good points you mentioned, we have included a paragraph in Discussion section to point these out.

We acknowledge the challenges associated with long pegRNAs, including susceptibility to exonuclease degradation, propensity to form secondary structures, and difficulties in chemical synthesis. These issues are common to the EXPERT as well as other two-pegRNAs systems. Indeed, when compared to the pegRNA of twinPE, the EXPERT's ext-pegRNA is approximately 15-bp longer (HS length). We want to point out however, unlike twinPE, which requires two long pegRNAs, EXPERT achieves editing with just one pegRNA and a regular sgRNA; whereas in another two-pegRNAs system PRIME-Del, the pegRNA length is typically around 30-bp for HS which is longer than the ext-pegRNA used in EXPERT (see below). Nevertheless, we are confident that technology advancement in RNA synthesis will help address this challenge. With regard to susceptibility to exonuclease degradation, this can be mitigated by incorporating protective motifs (Liu, Y. et al. 2021, Nelson, J.W. et al. 2021 and Li, X. et al. 2022), whereas the propensity of longer pegRNAs forming secondary structures can be alleviated through introduction of mismatches (Fig. 2 and supplementary Fig. 4) (Chen, P.J. et al. 2021 and Li, X. et al. 2022). These strategies may collectively enhance gene editing efficiency.

Comparison of pegRNAs between PRIME-Del, twinPE, and EXPERT

Specifically we have revised the following:

Lines 522-532: “We want to point out some future research directions. There are universal challenges associated with the use of long pegRNAs, including susceptibility to exonuclease degradation, propensity to form secondary structures (especially for small edits), and difficulties in chemical synthesis. This is especially the case for EXPERT because compared to the pegRNA used in twinPE, the ext-pegRNA in EXPERT is approximately 15-bp longer. The propensity of longer pegRNAs forming secondary structures may be alleviated through introduction of mismatches which have been attempted and proven effective^{5, 28}, whereas

incorporating protective motifs into the pegRNA will help to address the susceptibility to exonuclease degradation problem as suggested by some studies⁵⁻⁷, which should be tested to further improve the EXPERT system.” has been added.

R1.7: In figure 4 and 5, the comparison of EXPRT to PE2 needs to be carefully devised to avoid unfairness. In figure 4a authors uses ups-sgRNA target site directly as pegRNA target sequence. If that is the case, authors should also perform the comparison using the optimal pegRNA from PE2 and PE3 edits shown in Anzalone et al., Nature 2019 (Figure 2-5) and find a downstream DNA target for ext-pegRNA. This can help to determine whether ext-pegRNA’s improvement on editing % can be generally observed. Alternatively, if authors must pick the sites that is more frequently used for EXPERT, at least you should screen a few upstream protospacer sequence, optimize PBS and RTT for canonical PE, which is the recommended procedure in the Anzalone et al., Nature 2019 and other recently published papers on experimental and computational design of pegRNA.

Response: Thank you for your feedback. Based on your suggestions, we have supplemented the experiment by performing additional comparisons using the optimal pegRNAs from PE2 and PE3 edits as shown in Anzalone et al., Nature 2019 (including *HEK4*, *HEXA*, *VEGFA*). The results showed that EXPERT continues to demonstrate superior performance over PE2 at these sites, consistent with the trends observed in the manuscript data (Fig. 4a and Supplementary Fig. 8). We have incorporated these comparative results into the Results section.

Fig. 4a Frequencies of intended editing and indels introduced by PE2 and EXPERT at multiple loci. Bars represent the mean of n = 3 independent replicates.

Supplementary Fig. 9 Frequencies of intended editing and indels introduced by PE2 and EXPERT at multiple loci. Bars represent the mean of n = 3 independent replicates.

Reviewer #2 (R2):

R2 general Comments: The manuscript reports on modifications of prime editing technology that the authors find can improve editing performance in their studies. The modifications were rigorously tested using a variety of cell lines and targets. Overall the manuscript was clear and easy to interpret and understand.

Response: Thank you for your comments.

Major comments:

R2.1: The use of the word “replication” could use further clarification. Are the replicates shown from exactly the same experiment (same cells, same transfection, same day, same harvest times), or are they from completely independent experiments conducted on three separate days. In other words, are they technical or biological replicates? In many cases the error bars are extremely small, and so this is important to clarify.

Response: Thank you for pointing this out. These are biological replicates.

We have updated the following:

Lines 632-633: “At least three biological replicates were conducted per experiment to ensure the reliability of the data.” has been added.

R2.2: Lines 113-115. It seems as if this concept would be better placed after an introduction to the technology (after Figure 1 discussion?). There are a lot of studies described in Supplementary Figure 1. Perhaps they deserved their own paragraph of discussion to walk the reader through these studies.

Response: Thank you for your valuable suggestions. In response, we have integrated the research content and results from Supplementary Figure 1 (now is Supplementary Figure 2) into the section that follows the introduction of the technology.

Specifically we have revised the following:

Lines 119-156: “Both *cis* nicks and the upstream binding are essential for the differential editing capacity of the EXPERT system vs the canonical PEs. We first confirmed the role of the two *cis* nicks in the EXPERT system, by comparing it with the two EXPERT variants, EXPERT-a and EXPERT-b, that only generate one nick by using only the ups-sgRNA or only the ext-pegRNA, respectively (Supplementary Fig. 2a). We used them to edit the *HEK4_2* locus for introducing a 40-bp sequence replacement in HEK293T cells (Supplementary Fig. 2b). As expected, EXPERT (with two *cis* nicks) achieved 6.1% efficiency, in contrast to the 2.84% and below 0.1% efficiencies achieved by EXPERT-a and EXPERT-b (the single nick systems), respectively. Mechanistically, we speculate that this is because EXPERT generates two *cis* nicks that enhances the detachment of the original single-stranded DNA

fragment from the genome, thereby promoting subsequent processes (Supplementary Fig. 2c). These results confirmed the importance of the two *cis* nicks for the EXPERT system.

We next worked to confirm the role of the upstream binding. We included several PE2 variants in the experiment (Supplementary Fig. 2a). (i) PE2: it generates one nick by using the ups-sgRNA as its pegRNA. It does not have the upstream binding. (ii) PE2-a: it generates one nick by using the ups-sgRNA as its pegRNA. It does not have the upstream binding. We included a truncated ext-pegRNA in PE2-a although it does not create a 2nd nick. (iii) PE2-b: it generates two *cis* nicks, one using the ups-sgRNA, and another by using the ext-pegRNA. PE2-b also does not have the upstream binding. We compared these three PE2 variants with EXPERT, which has the upstream binding, to edit the same *HEK4_2* locus for introducing a 40-bp sequence replacement in HEK293T cells (Supplementary Fig. 2b). All these PE2 variants lacking the upstream binding design, regardless of the design to generate one nick or two *cis* nicks, achieved low efficient edits, ranging from 0.05% to 0.37%, consistent with the knowledge that current PEs are inefficient for long fragment edits. Remarkably, the efficiency achieved by EXPERT is 6.1%, 122.1-fold higher than that by PE2. These results confirmed the essential role of the upstream binding for the EXPERT system.

One consideration with the generation of two nicks in the EXPERT system is whether this will increase the unintended indel rates at the on-target site. Our results in the above experiments suggest that the presence of two *cis* nicks does not increase the likelihood of indel events in comparison to the one nick system PE2. The indel rate by EXPERT (using ext-pegRNA and ups-sgRNA) at the *HEK4_2* site is 0.28%, comparable to that by PE2 (with 0.2% indels) using the same ups-sgRNA (Supplementary Fig. 2b).

In summary, by introducing an extra upstream guide RNA (ups-sgRNA) to create an additional *cis* nick and an extended pegRNA (ext-pegRNA) that contains an upstream binding sequence, we construct a new PE tool EXPERT.” has been added.

Minor:

R2.3: The editors may not allow for reference to figures in the introduction.

Response: Thank you for your suggestions. We now removed references to figures in the Introduction.

We have updated the following:

Line 58: “(Fig. 1a)” has been removed.

R2.4: Abbreviations might need to be defined better upon first use.

Response: Thank you for your suggestions. We carefully reviewed our revised

manuscript and ensured that all abbreviations are defined upon their first use.

R2.5: Line 72. It might be helpful to provide the reported editing efficiencies for the available PEs described, just to provide context for the current work.

Response: Thank you for your professional suggestions. To provide context for the current work, we have provided the reported works of current PEs.

Specifically we have revised the following:

Lines 75-79: “Nevertheless, while many PE systems could achieve double digit (i.e., >10%) efficiencies in human cells, they can only do so within approximately a 10-bp range downstream of the nick^{1,11}. Hence there is a need for further expanding the editable region of PE systems, particularly for editing sites located more than 10-bp away downstream from the nick, and editing sites located upstream from the nick.” has been added.

R2.6: Line 91. “than that by” should be “compared to”.

Response: Thank you for your suggestion. We have revised the text in accordance with your advice.

We have updated the following:

Line 97: “than that by” has been corrected to “we compared EXPERT with.....”

R2.7: Figure 1. The editable portion is in “green” in the canonical PE side of Figure 1a. Should you make the upstream portion green as well in the EXPERT side. This would help to highlight that the editable portion of the target sequence has expanded (or at least, don’t keep both as red). Can you define the blue and purple and light blue structures in this figure? Figure 1b legend is not clear enough (what is being represented “respectively?”). Are you saying the canonical PE cannot edit, but the EXPERT will edit at two sites? What is RTT in this figure?

Response: Thank you for your suggestion.

R2.7.1 For comments about the colors used in the Fig. 1a, We apologize for the confusion caused by the colors in our previous figure. We have revised Figure 1 based on your suggestion. We have adjusted the color scheme to denote the expanded editable portion in the upstream region of the EXPERT side.

R2.7.2 Regarding the suggestion about the definitions of the blue, purple, and light blue structures in this figure, we have updated the figure to include these definitions. Specifically, the blue structure represents RT (reverse transcriptase), the purple structure (now changed to light blue) and the light blue structure (now changed to light gray) both represent nCas9 (Cas9 nickase, H840A). These definitions have been added to the figure legend.

R2.7.3 Regarding the comments on the Figure 1b legend, we apologize for any lack of clarity. Your understanding is correct: canonical PE cannot perform these two different edits, whereas EXPERT can do. We have revised the Figure 1b legend in the revised manuscript.

R2.7.4 In response to the comment “What is RTT in this figure ?”, RTT refers to the reverse transcriptase template, which consists of edit sequences (ES) and homology sequences (HS). Additionally, we have included a supplementary figure for further clarification on RTT (Supplementary Fig. 1).

Fig. 1 EXPERT expands the editing region of canonical PEs. **a** Schematics of canonical PEs and EXPERT systems. The deep blue thick-lined area (downstream region of the pegRNA nick) represents the editable region for canonical PEs and EXPERT. On the other hand, the green thick-lined area (upstream region of the pegRNA nick) is not editable by canonical PEs, whereas the region marked by the light blue thick-lined area can be edited by EXPERT. nCas9, Cas9 nickase (H840A). RT, reverse transcriptase. **b** Frequencies of intended editing and indels were performed by EXPERT for two different edits, which canonical PEs cannot perform. PBS, primer binding site. RTT, reverse transcriptase template. Bars represent the mean of n = 3 independent replicates. **c** EXPERT performs simultaneous editing of both sides regions of the pegRNA nick at the *VEGFA* site. Bars represent the mean of n = 3 independent replicates.

Supplementary Fig. 1 The structural components of ext-pegRNA.

R2.8: It is not clear why VEGFA was used in Figure 1. Was there some type of rationale to utilize this target sequence from the beginning? It is not critical to know, but would help to explain the experiment conducted in context.

Response: We chose the *VEGFA* (now referred to as *VEGFA_1*) site because it is a commonly used target in gene editing research, offering multiple NGG sequences on the same strand, which facilitates the design of ext-pegRNA and ups-sgRNA. In response to your suggestion, we have included additional explanations in the Results section.

Specifically we have revised the following:

Lines 166-168: “To test the effectiveness of EXPERT, we first conducted experiments to evaluate its capability in generating two different edits at the *VEGFA* site.” has been corrected to “The *VEGFA_1* site is chosen because it provides multiple NGG sequences on the same strand, thereby facilitating the design of ext-pegRNA and ups-sgRNA.”

R2.9: It never became completely clear if two guide RNAs are expressed from the vectors. Could you expand a bit on exactly what the plasmid vectors look like in PE vs EXPERT-PE?

Response: We apologize for not clearly explaining. The two guide RNAs (i.e., ext-pegRNA and ups-sgRNA) are expressed in one vector.

The main difference of plasmid vectors between PE and EXPERT is that the EXPERT vector includes an additional guide RNA expression module. We have included a description and schematic information of the EXPERT plasmid used in the Methods section and Supplementary Fig. 17.

Specifically we have revised the following:

Lines 556-561: “Three versions of EXPERT vectors were utilized in this study. Schematic diagrams depicting the structure of these vectors are presented in Supplementary Fig. 17. Of note, both the ext-pegRNA and ups-sgRNA are expressed in the same one vector in all versions. The main difference of plasmid vectors between PE and EXPERT is that the EXPERT vector includes an additional guide RNA expression module.” has been added.

R2.10: Line 674. “that using” should be “that used”.

Response: Thank you for your suggestion. We have revised the text.

We have updated the following:

Line 811: “that using” has been corrected to “that used”

R2.11: Line 682. As the authors have done for “Figure 2c” it would be helpful to the reader if the consequences of the hybridization of the 5’ flap are explained.

Response: Thank you for your suggestion. We have now included explanatory notes in the Figure 2f (which is now Supplementary Fig. 4a), similar to what was done in Figure 2c.

Supplementary Fig. 4 Enhancing EXPERT editing efficiency through the introduction of mismatches on ext-pegRNA. **a** Diagram of the hybridization of the 5’ Flap of the original DNA strand to complementary ext-pegRNA. The hybridization can also hinder the reverse transcription process of RT enzymes, leading to editing failure. **b** The pattern of introducing mismatch on ext-pegRNA region that hybridizes with the 5’ Flap of the original DNA strand. **c** Frequencies of intended editing and indels were performed by EXPERT at the indicated target sites under the different number of mismatches on ext-pegRNA region that hybridizes with the 5’ Flap (top). The product purity (intended edits: indels ratio) of EXPERT that introducing different number of mismatches (bottom). Bars represent the mean of n = 3 independent replicates.

R2.12: Lines 163-168. The text is a bit confusing as it is not really obvious from the figure that that intended site is “away” from the nick site. Was this site chosen because the frequency was very low, thus you could then see a signal when you added the mismatch?

Response: We apologize for the confusion. We have revised the figure to clearly illustrate the distance between the editing region and the nick site (m) (Fig. 2c).

You are right about the reason for choosing this site. When we first attempted EXPERT, we observed low efficiency for edits when the editing region is more than 14 nt away from the nick, whereas edits on the region close to the nick achieved high efficiency (Supplementary Fig. 17c). Therefore, we hypothesized that when the editing region is away from the nick, the homologous hybridization might have affected the synthesis of the RT enzyme (Fig. 2c). Indeed, by introducing mismatches as a means to prevent homologous hybridization, we see an increase in the editing rates for regions that are away from the nick (Fig. 2d and 2e).

Fig. 2c Diagram of the hybridization of the 3' Flap of the original DNA strand to complementary ext-pegRNA. The hybridization can hinder the reverse transcription process of RT enzymes, leading to editing failure. The bold red line area denotes the editing region. “m” indicates the distance between the editing region and ext-pegRNA nick.

R2.13: Line 259-263. While there was no difference between the experimental groups, each showed some gRNA-independent off target effects? How many were seen? Is this still a problem for the technology? It seems as if the EXPERT system did not reduce the number of off target effects.

Response: Thank you for your feedback. You are right, the EXPERT system did not reduce the number of off target effects; nor did it increase the number of off target edits.

R2.13.1 For comments about “While there was no difference between the experimental groups, each showed some gRNA-independent off target effects ? How many were seen ?”, we want to clarify that, we conducted off-target analysis following the reported methods using whole-genome sequencing (Runze Gao. et al. 2022). Wild type cells, non-target cells and EXPERT-edited cells were compared separately against the reference genome to obtain the number of variations.

Due to inherent SNPs and indels (relative reference genome) in the wild type cells, this group naturally exhibited some variations (number of SNPs: 3, 528, 399, number of indels: 1, 141, 410). Upon analysis, comparing the non-target group (number of SNPs: 3, 529, 559, number of indels: 1, 150, 181) and EXPERT-edited group (*EMXI*-number of SNPs: 3, 524, 121, *EMXI*-number of indels: 1, 125, 381; *VEGFA_1*-number of SNPs: 3, 523, 318, *VEGFA_1*-number of indels: 1, 131, 086)

with the wild type control, we found no significant increase in off-target mutations attributable to EXPERT (Supplementary table 4). The number of SNPs and indels in the *EMXI* and *VEGFA_1* groups (edited) are actually lower than those in the non-target group. Therefore, we concluded that EXPERT did not increase gRNA-independent off-target effects.

R2.13.2 For comments about “Is this still a problem for the technology?”, while our work show that EXPERT does not increase the gRNA independent off-target editing events, it should be noted that nor does it reduce the incidence of such event, a remaining challenge to be addressed in PE research. We have included a discussion in the revised manuscript.

Specifically we have revised the following:

Lines 532-535: “It also should be noted that the EXPERT system does not reduce the gRNA-independent OT effects, although it does not increase the OT effects either. How to minimize gRNA-independent OT effects is a general challenge for PE research which remains to be addressed.” has been added.

R2.14: Figure 4 legend title does not reflect the various aspects of the figure, especially the off-target effects. Should there be a broader title chosen?

Response: Thank you for your suggestion. We have updated the legend title for Figure 4 to encompass a broader scope, including off-target effects and other relevant aspects of the figure.

Specifically we have revised the following:

Lines 837-838: “The EXPERT strategy enhances the editing efficiency of most PE variants while maintaining low indel rate.” has been corrected to “The EXPERT strategy enhances editing efficiency with high product purity and low off-target effects.”

Reviewer #3 (R3):

R3 general comments: This is potentially an interesting and novel adaptation of Prime Editing called EXPERT. Briefly, it builds on canonical PE (also known as PE2) by adding an extra guide RNA and an “extra” elongated 3’ end of the pegRNA (I say “extra” elongated pegRNA as the pegRNA is itself already an elongation of the 3’ end of a standard CRISPR guide RNA).

The consequence of positioning the gRNA upstream of the nick site created by PE2/pegRNA is that an additional 3’ flap is created which provides an alternate primer binding site (PBS). And since the “extra” elongated pegRNA has a PBS to bind this additional 3’ flap, that’s where the editing starts.

Not to diminish the novelty and work presented herein, this isn’t a whole new system, rather it’s still PE2 but with a nicking gRNA in a different position to the PE3 system, and an “extra” elongated pegRNA. This does allow the creation of longer edits than previously reported, and that in itself is a very important development. As authors point out, the use of the gRNA to nick on the same strand (a *cis* site), is important in reducing the likelihood of indels. In many cases, this appears to work very well, but as mentioned below, there are few examples, where I’m not convinced (yet) that this is always the case. But it’s a big improvement.

Response: Thank for your valuable feedback. Please see our responses to your specific comments below.

My specific comments are as follows.

Abstract

R3.1: Ln 36 – insert the word “region” after “targeting the”

Response: Thank you for your suggestion. We have revised the text according to your advice.

Specifically we have revised the following:

Line 36: The word “region” has been inserted.

R3.2: Ln 44 – authors should include a statement explaining their idea why *cis* nicking should reduce indels

Response: Thank you for your suggestion. We have revised the text in the Abstract and included a statement in the Discussion explaining why *cis* nicks dose not increase indels.

Cis nicks are two “cuts” on the same strand of DNA (i.e., not double strand breaks), therefore in theory can be repaired precisely using the intact strand template (Hyodo,

T. et al 2020). In contrast, *trans* nicks are two “cuts” with each on one strand therefore constitutes a DSB, often leading to indels after NHEJ repair. This explanatory content has been included in the revised manuscript.

Specifically we have revised the following:

Lines 43-45: “Safety wise, the use of ups-sgRNA does not increase the undesirable insertion and deletion (indel) rates at the target site;” has been corrected to “Safety wise, the use of ups-sgRNA does not increase the rates of undesirable insertions and deletions (indels) because the two nicks are on the same strand;”

Intro (minor comments unless otherwise stated)

R3.3: Ln 49 – insert “*Strep. pyogenes*” before “Cas9” (they mention source of RT on Ln 50 so should include source of Cas9).

Response: Thank you for your suggestion. We have revised the text according to your advice.

Specifically we have revised the following:

Lines 48-49: The word “*Strep. pyogenes*” has been inserted before “Cas9”.

R3.4: Ln51-55 – may be better as two sentences. First part is wrong; as authors know, the pegRNA/Cas9-RT creates a nick which liberates a 3’ flap which binds the PBS on the 3’ end of the pegRNA. They should also state the template part of the pegRNA is known as the RTT or edit sequence.

Response: Thank you for your suggestion. Based on your feedback, we have revised this section.

Specifically we have revised the following:

Lines 50-55: “To enable editing, a prime editing guide RNA (pegRNA) is used, whose 3’ extension will bind to the primer binding site (PBS) on the non-targeted strand of the pegRNA, serving as the starting point for the reverse transcription (RT) of the new DNA strand by using part of the pegRNA as the template” has been corrected to “To enable editing, a prime editing guide RNA (pegRNA) is used, which includes a protospacer defining the target site, a sgRNA scaffold, and a 3’ extension encoding the desired edit. This extension contains a primer binding site (PBS) complementary to a segment of the DNA protospacer, as well as an RT template (RTT) that encodes the desired edit and genomic homologous sequences¹”

R3.5: Ln 60 – I don’t think it expands the scope, rather it specifically extends the length of edits that can be done by PE. To justify this request, PE extended the scope of editing relative to BE; BE could only do C>T or A>G, whereas PE could do all 12

SNP changes plus inserts and deletions. Thus, EXPERT doesn't increase the scope, just the length.

Response: Thank you for your insightful observation. We agree and have revised all occurrences of "scope" to "region" throughout the entire manuscript.

R3.6: Ln 62 – TJ-PE was INSPIRED by the activity of endogenous retrotransposons, but to best of my knowledge uses the Cas9/RT prime editor – it does not "harness the activity of endogenous retrotransposons"

Response: We apologize for the confusion. As you correctly pointed out, while TJ-PE was inspired by the activity of endogenous retrotransposons, it employs the nCas9/RT prime editor and does not directly harness the activity of endogenous retrotransposons. Following your suggestion, we have revised the sentence.

Specifically we have revised the following:

Lines 61-64: "The development of CPE³ and TJ-PE⁴, which respectively involve combining Cas12a with circular RNA and harnessing the activity of endogenous retrotransposons, also broadens the editing scope." has been corrected to "The circular RNA-mediated prime editor (CPE) which combines Cas12a with circular RNA³, and template-jumping PE (TJ-PE)⁴, inspired by the genomic insertion mechanism of retrotransposons, also expand the editing region."

R3.7: Ln67-68 – edit slightly, as written it's literally incorrect. E.g. insert "alter" or "change" before "certain amino residues"

Response: Thank you for your suggestion. We have revised the text according to your advice.

Specifically we have revised the following:

Line 70: The word "alter" has been inserted before "certain amino acid residue".

R3.8: Ln71-75 – this is an important point to be considered. EXPERT is an interesting system, but I would invite the authors to rephrase this slightly; I'll phrase my point as carefully as I can. I don't think EXPERT can edit upstream, rather it's shifting the start site upstream so the overall effect is a longer editing tract. I like this strategy, but I think it's slightly incorrect as written.

Response: Thank you for your suggestion. We agree with your point that EXPERT shifts the start site upstream and the overall effect of EXPERT is a longer editing tract. We have revised this sentence.

Specifically we have revised the following:

Lines 75-79: "Nevertheless, the editing efficiency of PEs remains to be improved, and to be best of our knowledge, all existing PEs are still limited to editing the

downstream genomic region of the pegRNA nick such that the upstream genomic region of the pegRNA nick remains uneditable.” has been corrected to “Nevertheless, while many PE systems could achieve double digit (i.e., >10%) efficiencies in human cells, they can only do so within approximately a 10-bp range downstream of the nick^{1, 11}. Hence there is a need for further expanding the editable region of PE systems, particularly for editing sites located more than 10-bp away downstream from the nick, and editing sites located upstream from the nick.”

R3.9: LN78-80 – just a comment, this is a good point well made, and potentially an important feature of EXPERT relative to most (all?) other PE systems to date.

Response: Thank you for your comment. It highlights an important feature of EXPERT compared to most other PE systems (all two-pegRNAs systems) to date.

R3.10: LN 91-92 – whilst they have a specific example of an increase of 122-fold relative to PE2, I don't think that's a completely fair representation as many times, the increase is no where near this. Maybe, stating an average and a range, e.g. SEM or 95% confidence level etc. would better portray the increase in activity. Additionally, it may be fairer to compare the fold increase with the PE3 values which also uses a second gRNA, although I would concede that PE3 uses the nicking guide to boost efficiency in a potentially different way.

Response: Thank you for your suggestion. Please see below.

R3.10.1 For the comments about “that’s not a completely fair representation as many times”, we agree with you. Based on your suggestion, we have incorporated average efficiency enhancement in the Abstract section and revised this description in the Introduction section.

R3.10.2 For the comments about “it may be fairer to compare the fold increase with the PE3”, we have conducted the comparison between the EXPERTmax strategy and PE3max (Fig. 5a, as shown in the red box). We also have included a supplementary figure to compare the overall efficiency of EXPERTmax and PE3max (Supplementary Fig. 13). The results demonstrated that EXPERTmax outperformed PE3max in editing rate by an average of 1.7-fold. We have added relevant results in the revision.

Fig. 5a Frequencies of intended editing and indels introduced by PE2max, PE3max, PE4max, PE5max and their corresponding EXPERTmax systems at endogenous sites. Bars represent the mean of n = 3 independent replicates.

Supplementary Fig. 13 Statistical analysis of normalized editing frequencies of EXPERTmax. Setting the frequencies induced by PE3max as 1. n = 4 editing from three independent experiments shown in Fig. 5a, respectively.

Specifically we have revised the following:

Line 42: “with an average improvement of 3.12-fold” has been added.

Lines 97-102: “Additionally, we demonstrate that the EXPERT strategy improves the editing efficiency, up to 122 times than that by PE2, while maintaining a low indel rate at the target site and low off-target edit rate genome wide” has been corrected to “We compared EXPERT with representative single-pegRNA system PE2 and two-pegRNAs system twinPE and demonstrated that EXPERT achieves higher product purity. The results reveal that EXPERT can efficiently perform various types of prime editing on both sides of the ext-pegRNA nick, a task that is unattainable by canonical PEs, while maintaining low indel rates at the target site and minimal off-target effects genome-wide.”

Lines 402-403: “Without including the additional nicking sgRNA, EXPERTmax had an average 1.7-fold higher editing efficiency than PE3max (Supplementary Fig. 13).” has been added.

R3.11: Ln 92-93 – again, to be strictly correct, this isn’t really changing the function of PE, and it’s not really changing the versatility, but it does increase the length of sequences that can be corrected.

Response: Thank you again for your suggestion. Following your advice, we have revised this sentence in the Abstract and Introduction.

Specifically we have revised the following:

Lines 45-46: “Our work introduces EXPERT to the toolbox of gene editing, which is expected to improve the versatility and function of PEs.” has been corrected to “Our

work introduces EXPERT as a novel PE tool with significant potential in life sciences.”

Lines 103-105: “Collectively, EXPERT represents a novel and effective gene-editing tool that complements the PE toolbox and holds significant value in life sciences.” has been revised.

R3.12: Ln106/Figure 1b – the color scheme in 1b doesn’t match the color scheme in 1a – i.e. 1a shows upstream edits in red, but 1b shows the first edit in green, but it’s a upstream edit. Then the next upstream edit is in blue! Maybe both should be in red, one full color, the other with a striped pattern maybe?

Also, would a red triangle to shown the nick site help – most papers use that symbol?

Response: Thank you for your feedback. We apologize for the inconsistency in the color scheme between Figure 1a and 1b, which may have caused confusion. Based on your suggestions, we have revised the color scheme in Figure 1a and 1b accordingly. We have now used green for upstream edits, distinguishing them clearly with different full color. Additionally, we have included a red triangle to denote the nick site, aligning with common practices in the field.

Fig. 1 EXPERT expands the editing region of canonical PEs. **a** Schematics of canonical PEs and EXPERT systems. The deep blue thick-lined area (downstream region of the pegRNA nick) represents the editable region for canonical PEs and EXPERT. On the other hand, the green thick-lined area (upstream region of the pegRNA nick) is not editable by canonical PEs, whereas the region marked by the light blue thick-lined area can be edited by EXPERT. nCas9, Cas9

nickase (H840A). RT, reverse transcriptase. **b** Frequencies of intended editing and indels were performed by EXPERT for two different edits, which canonical PEs cannot perform. PBS, primer binding site. RTT, reverse transcriptase template. Bars represent the mean of n = 3 independent replicates. **c** EXPERT performs simultaneous editing of both sides regions of the pegRNA nick at the *VEGFA* site. Bars represent the mean of n = 3 independent replicates.

R3.13: Ln 109 – “In the work” doesn’t make sense. Do they mean, “In this study…” or “In our strategy …“?

Response: Thank you for your suggestion. We have removed this phrase in the revision.

R3.14: Ln118-125 and Figure 1 – not clear from the figure or the text if the two edits are performed by two different pegRNAs, or if one peg can perform both edits – can this be clarified please.

Response: Thank you for your feedback. We apologize for the lack of clarity. We meant that the two edits can be accomplished by the same ext-pegRNA spacer but different 3’ extension. Based on your suggestion, we have included the explanation in the manuscript.

Specifically we have revised the following:

Lines 159-174: “Canonical PEs are unable to edit the upstream region of the pegRNA nick. The EXPERT system, thanks to the introduction of two *cis* nicks and the use of ext-pegRNA, in theory should be able to edit that unreachable region by canonical PEs.

To validate this hypothesis, we performed two edits at the *VEGFA_1* site, both located in the upstream region of the ext-pegRNA nick: (i) replacing a 37-bp sequence (*VEGFA_1* -37to-1 replace 37bp); (ii) deleting a 37-bp sequence (*VEGFA_1* -37to-1 del 37bp) (Fig. 1b). The *VEGFA_1* site is chosen because it provides multiple NGG sequences on the same strand, thereby facilitating the design of ext-pegRNA and ups-sgRNA. For the first edit, a high replacement efficiency at 33.7% with low indel rate (0.52%) is achieved. Similarly, for the second edit, the precise deletion rate is high at 18.8% with a low indel rate at 0.8%. We also attempted a third edit at this site: (iii) *VEGFA_1* -37to-1 replace extended loxp and +1 TtoC, to test whether EXPERT allows for simultaneous editing of both sides of the ext-pegRNA nick. The results showed that the efficiency of this simultaneous editing on both sides of the ext-pegRNA nick has reached 10.7%, with a low indel rate at 0.39% (Fig. 1c).” has been added.

R3.15: Ln126 – the prime editing field suffers from a challenge of nomenclature with PE being used to refer to both the process (prime editing) and the editors (prime editors) – thus I struggle with the first sentence. There is no different between the prime editor used in canonical prime editing, and expert – the difference is one uses

only a pegRNA whereas the other uses an “extra’ elongated pegRNA and a nicking gRNA. Simplest solution could be to replace “canonical PE” with “canonical prime editing strategies”?

Response: Thank you for your suggestion. In the revised manuscript, “PEs” refers to “prime editing systems”. We have updated the text throughout.

Specifically we have revised the following:

Line 31: “Prime editors (PEs)” has been corrected to “Prime editing systems (PEs)”

Line 48: “The canonical Prime editor (PE) ” has been corrected to “The initial prime editing (PE) system”

R3.16: Ln 135 – insufficient data (yet) to support “proves” so use “ supports the hypothesis” instead.

Response: Thank you for your suggestion. We agree with your point and have revised our manuscript accordingly.

Specifically we have revised the following:

Line 177: “proves” has been corrected to “supports the hypothesis”

R3.17: Ln 147 – I agree the highest is achieved at 38-51 nt, but it’s only 2-4% so this should also be mentioned in the text as well as the figure. Authors claim earlier a 122-fold improvement, but in many cases it’s much lower, so I think referring to the % activity in the text as they go a long is important, and will also help in finding a figure that better reflects the overall fold improvement of EXPERT.

Response: Thank you for your suggestion. We apologize for not clarifying this in our manuscript. We have revised our manuscript accordingly.

Specifically we have revised the following:

Lines 187-190: “The results showed that mCherry signals were detectable when the DCN is between 23-126nt, and the highest efficiency was achieved when DCN is between 38 and 51nt.” has been corrected to “The results showed that the mCherry signals are detectable when the DCNs are between 38-96 nt, but more robust when DNCs are between 38-71 nt with efficiencies ranging from 1.75% to 7.44%.”

R3.18: Ln 148-152 – as another general comment, they need to comment on indels – they are clearly present, in some cases at quite high levels, but in the abstract, intro and discussion, indels are described as very low – the data in the text doesn’t always support these “blanket” statements about the low levels of indels in other parts of the manuscript.

Response: Thank you for your suggestion. We have included the description of indels

in this section accordingly. And we have also revised our text regarding indels in the manuscript (including Abstract, Intro and Discussion).

Specifically we have revised the following:

Lines 192-196: “The results showed that the editing rates peaked when DCN is between 32 to 40 nt (Fig. 2b). At the *VEGFA_1* locus, the highest editing rate of 33.7% with an indel rate of 0.52% is achieved when DCN is 37 nt long. At the *HEK4_1* locus, the highest editing rate of 11.9% with an indel rate of 2.23% is achieved when DCN is 32 nt long.” has been added.

R3.19: Can they also comment why the editing with 24nt spacing is so low – is it simply a steric hindrance issue?

Response: Thank you for your suggestion. The DCN of 24nt at this site (*HEK4_1*) almost completely abolished any intended editing, which is similarly observed in the mCherry reporter cells: when a 23 nt DCN is used the editing rate is much lower at 0.3%. So we speculate that these observations suggest that DCNs of 24 nt or shorter might be a threshold number that causes steric hindrance between two adjacent nCas9 proteins. Our results are consistent with previous studies that have reported using double nicking for gene editing (F. Ann Ran et al. 2013 and Bin Shen et al. 2014). In response, we have included relevant details in the Results section.

Specifically we have revised the following:

Lines 196-203: “Interestingly, in the edit at this site (*HEK4_1*), a slightly shorter DCN of 24 nt almost completely abolished any intended editing. This is similarly observed in the mCherry reporter cells: when a 23 nt DCN is used the editing rate is only 0.3% but when a 38 nt DCN is used the editing rate reached a peak at 7.44% (Fig. 2a). We speculate that these observations suggest that a DCN of 24 nt or shorter might cause steric hindrance between two adjacent nCas9 proteins; in consideration of this, DCNs of 24 nt or shorter should be avoided in the EXPERT system.” has been added.

R3.20: Ln 156-162 – this concept is nicely introduced – it’s an intriguing concept.

Response: Thank you for comments.

R3.21: Ln 170-183 – the VEGF and HEK4 editing profiles and indel profiles are quite different – any idea why, and they don’t really comment upon this – they should. For example, VEGF has almost no indels, but HEK4 has lots. Also, why do they show a “purity profile” in Fig2h, but not Fig 2e?

Response: Thank you for your comments. Please see below.

R3.21.1 For comments about “why the VEGF and HEK4 editing profiles and indel profiles are quite different”, we actually don’t know except that we know gene editing outcome are often sequence dependent. Indeed, in the first publication

reporting PE2 and PE3, editing at the *VEGFA* and *HEK4* locus also resulted in quite different editing profiles and indel profiles, similar to our results (Anzalone et al Nature 2019). Another evidence to show subtle sequence difference lead to substantial difference in editing outcome including the indel rate can also be found in the current work, where introducing different numbers of mismatches at the same locus (*HEK4_I*) leads to significant variations in both editing profiles and indel profiles (Fig. 2 and Supplementary Fig. 4). Therefore, we infer that different loci and editing sequences lead to variations in editing profiles and indel profiles. At a rather high level perspective, the indel rate that we observe reflects a balance of cut, uncut, precise repair, and unprecise repair. Any of these four events could be affected in a sequence dependent manner and each could contribute to the final “indel” rate (as well as wild-type and precisely edited rates). But exactly how the sequence context between *VEGFA* and *HEK4* sites affect these four events at the molecule level remain elusive, and in fact beyond our capacity. We have included a paragraph of discussion in the revision.

R3.21.2 For comments about “why do they show a ‘purity profile’ in Fig2h, but not Fig 2e ?”, we apologize for this oversight. In response, we have included purity analysis for Fig 2e in our revised manuscript to provide a comprehensive view of our findings (Fig. 2f).

Fig. 2 Enhancing EXPERT editing efficiency through optimizing the distance between ups-sgRNA and ext-pegRNA, introducing mismatches on ext-pegRNA, or optimizing PBS length. a Schematic diagram of using ext-pegRNA and ups-sgRNA at different distances to

modify the stop codon in the 293T-reporter cell line to restore mCherry function (top). Frequencies of editing by EXPERT that used ext-pegRNA and ups-sgRNA at different distances were quantified by flow cytometry (bottom). Bars represent the mean of $n = 3$ independent replicates. **b** Frequencies of intended editing and indels by EXPERT that used ext-pegRNA and ups-sgRNA at different distances were quantified at *VEGFA_1* and *HEK4_1* sites. Bars represent the mean of $n = 3$ independent replicates. **c** Diagram of the hybridization of the 3' Flap of the original DNA strand to complementary ext-pegRNA. The hybridization can hinder the reverse transcription process of RT enzymes, leading to editing failure. The bold red line area denotes the editing region. "m" indicates the distance between the editing region and ext-pegRNA nick. **d** The pattern of introducing mismatch on ext-pegRNA region that hybridizes with the 3' Flap of the original DNA strand. **e** Frequencies of intended editing and indels were performed by EXPERT at the indicated target sites under the different number of mismatches on ext-pegRNA region that hybridizes with the 3' Flap. **f** The product purity (intended edits: indels ratio) of EXPERT that introducing different number of mismatches (bottom). **g** The impact of PBS length on editing efficiency at different sites. Bars represent the mean of $n = 3$ independent replicates.

Specifically we have revised the following:

Lines 256-261: "Moreover, EXPERT exhibits differential unintended indel rates at the *VEGFA_1* and *HEK4_1* loci, possibly attributable to the sequence characteristics of these sites or the editing sequence. It is important to note that incorporating appropriate mismatches can enhance editing efficiency and thereby reduce indel rates (Fig. 2e and Supplementary Fig. 4c)." has been added.

Lines 236-238: "After analyzing the purity of edited products, the results indicate that introducing mismatches can enhance product purity at both the *VEGFA_1* and *HEK4_1* sites (Fig. 2f)." has been added.

R3.22: Also, why such a high level of indels in Fig 2h, and why does it increase with the number of mismatches.

Response: Thank you for your feedback. In Fig 2h, our findings indicate that the indel rate varies depending on the number of mismatches introduced (We have relocated this data to **Supplementary Figure 4** due to the substantial increase in data in **Figure 2**). As we stated in response to **R3.21.1**, the indel rate that we observed is a balance of cut, uncut, precise repair and unprecise repair. When the "mismatch" number is low, the precise repair utilizing the template is low (likely due to the hybridization between the repair template and the target region sequence) thus the unprecise repair rate is high, resulting in observed high indel rate. On the other hand, when the mismatch reaches a threshold (e.g., 9 bp mismatches) such that the template-target hybridization is eliminated, the precise repair rate is increased whereas the unprecise repair is decreased, therefore presenting as very low indel rate. We have included a discussion on this point in the revision.

Specifically we have revised the following:

Lines 256-261: “Moreover, EXPERT exhibits differential unintended indel rates at the *VEGFA_1* and *HEK4_1* loci, possibly attributable to the sequence characteristics of these sites or the editing sequence. It is important to note that incorporating appropriate mismatches can enhance editing efficiency and thereby reduce indel rates (Fig. 2e and Supplementary Fig. 4c).” has been added.

R3.23: And, why did they do continuous tracts of mismatches – did they test replacing, for example, every second base?

Response: Thank you for your suggestions. According to your suggestion, we tested another strategy to introduce mismatches, i.e., a mismatch every *n* nucleotides, where *n* = 3 or 5. Interestingly, our findings indicate that the editing efficiency of ext-pegRNA with mismatches introduced every 3 or 5 nucleotides closely resembles that of achieved by ext-pegRNA with long consecutive mismatches (Supplementary Fig. 7). The data suggest that “1 in 5 mismatch” (as well as the “1 in 3 mismatch”) is a working strategy to for the design of ext-pegRNA in EXPERT. We have included the outcomes of these interval mismatch experiments in the Results section of our manuscript.

Supplementary Fig. 7 Frequencies of intended editing and indels were performed by EXPERT at the indicated target sites under the different number of mismatches on ext-pegRNA region. Bars represent the mean of *n* = 3 independent replicates.

Specifically we have revised the following:

Lines 251-256: “We also tested another strategy to introduce mismatches. Specifically, the mismatches are introduced at intervals of every 3 and 5 nucleotides, respectively (Supplementary Fig. 7). The results indicate that the editing efficiency of ext-pegRNA with such mismatches closely resembles that observed with ext-pegRNA featuring 11 consecutive mismatches. These results suggest an alternative mismatch design of the ext-pegRNA for achieving effective editing with EXPERT.” has been added.

R3.24: Ln208-209. Suggest minor re-write to emphasize that “helper gRNA” is a new term they are proposing. Was “To improve EXPERT’s capacity to insert or replace long fragment, we included a Helper gRNA positioned between the ups-sgRNA and ext-pegRNA (Fig. 3c)”.Suggest “ To improve EXPERT’s capacity to insert or replace

long fragment, we included an extra gRNA, positioned between the ups-sgRNA and ext-pegRNA, which we refer to as the Helper gRNA(Fig. 3c)".

Response: Thank you so much for your suggestion. We have revised the manuscript following your recommendation.

Specifically we have revised the following:

Lines 304-305: "To improve EXPERT's capacity to insert or replace long fragment, we included a Helper gRNA positioned between the ups-sgRNA and ext-pegRNA" has been changed to "To address this, we included an extra gRNA, positioned between the ups-sgRNA and ext-pegRNA, which we refer to as the Helper gRNA"

R3.25: Ln 210-213 – suggest minor reordering of sentence to better reflect the most likely real life order of events was, "...which would facilitate the detachment of the original DNA strand, and may recruit an additional nCas9-RT enzyme between the original two nicks thereby increasing the overall reverse transcription".

Suggest, "... which may recruit an additional nCas9-RT enzyme between the original two nicks, and then facilitate the detachment of the original DNA strand, and thereby increasing the overall reverse transcription".

Response: Thank you so much for your suggestion. We have revised the manuscript based on your advice.

Specifically we have revised the following:

Lines 307-309: "which would facilitate the detachment of the original DNA strand, and may recruit an additional nCas9-RT enzyme between the original two nicks thereby increasing the overall reverse transcription" has been corrected to "which may recruit an additional nCas9-RT enzyme between the original two nicks, and then facilitate the detachment of the original DNA strand, and thereby increasing the overall reverse transcription."

R3.26: Ln 220-223 the statement about indels is too strong regarding their prediction that nicking in *cis* "confirms" their speculation. Indeed, they have quite a few indels in Fig 2 and 3b. A nuanced version of this statement is suggested.

Response: Thank you for your feedback. We acknowledge that the statement regarding indels on lines 220-223 may have been too strong in asserting that nicking in *cis* "confirms" our speculation. In response to your suggestion, we have adjusted the wording from "confirm" to the more nuanced "indicate".

Specifically we have revised the following:

Line 317: "again confirming our speculation" has been corrected to "further indicating"

R3.27: Ln 235 – it might be useful to discuss why they chose to compare EXPERT with PE2 rather than PE3, since both PE3 and EXPERT use a pegRNA and nicking guide – it might have been a fairer comparison when looking at absolute levels of editing, and also indel formation.

Response: Thank you very much for your feedback. We apologize for any confusion in our manuscript. The reason we compared EXPERT with PE2 rather than PE3 is because EXPERT and PE2 both target cleavage of a single strand of DNA, whereas PE3 targets cleavage of both DNA strands. We speculated that, compared to PE2, EXPERT will enhance the editing efficiency and dose not increase the indel rates. We have clarified this rationale and included the comparison between EXPERT and PE2 in the Results section.

Specifically we have revised the following:

Lines 329-330: “We first compared the editing outcome and the indel rates between EXPERT and the single pegRNA system PE2, both of which induce nicks on a single DNA strand.” has been added.

R3.28: Figure 4a – this is a really nice set of data, but two comments. First, given this is the most comprehensive comparison of PE2 and EXPERT, I don’t see how this data leads them to the 122-fold increase in editing relative to PE2 claim in the abstract?

Response: We appreciate the feedback. The 122.1-fold increase in editing efficiency relative to PE2 is supported by data from Supplementary Fig. 2. We regret the oversight in not explicitly stating this in the main text earlier. To clarify, we have now included detailed explanations regarding Supplementary Fig. 2 in the Results section.

Supplementary Fig. 2 The additional *cis* nick and the upstream binding are necessary for improving the editing efficiency by EXPERT. a Schematic diagram of the composition of

several systems (EXPERT-a, EXPERT-b, EXPERT, PE2-a, PE2-b, PE2). These systems have different compositions and modifications. **b** These systems each performed the same replacement of a 40bp DNA fragment at the *HEK4_2* site. Bars represent the mean of n = 3 independent replicates. **c** Models for DNA 5' Flap replacement repair of canonical PEs and EXPERT intermediates. In the canonical PEs, if the Flap replacement fails, the non-edited strand will be used as the template to repair the genome; If the Flap replacement succeeds, the editing process will continue. In the EXPERT, an additional nick has been added to the original strand, making it easier for the original strand to detach from the genome, thereby promoting Flap replacement.

Specifically we have revised the following:

Lines 119-156: “Both *cis* nicks and the upstream binding are essential for the differential editing capacity of the EXPERT system vs the canonical PEs. We first confirmed the role of the two *cis* nicks in the EXPERT system, by comparing it with the two EXPERT variants, EXPERT-a and EXPERT-b, that only generate one nick by using only the ups-sgRNA or only the ext-pegRNA, respectively (Supplementary Fig. 2a). We used them to edit the *HEK4_2* locus for introducing a 40-bp sequence replacement in HEK293T cells (Supplementary Fig. 2b). As expected, EXPERT (with two *cis* nicks) achieved 6.1% efficiency, in contrast to the 2.84% and below 0.1% efficiencies achieved by EXPERT-a and EXPERT-b (the single nick systems), respectively. Mechanistically, we speculate that this is because EXPERT generates two *cis* nicks that enhances the detachment of the original single-stranded DNA fragment from the genome, thereby promoting subsequent processes (Supplementary Fig. 2c). These results confirmed the importance of the two *cis* nicks for the EXPERT system.

We next worked to confirm the role of the upstream binding. We included several PE2 variants in the experiment (Supplementary Fig. 2a). (i) PE2: it generates one nick by using the ups-sgRNA as its pegRNA. It does not have the upstream binding. (ii) PE2-a: it generates one nick by using the ups-sgRNA as its pegRNA. It does not have the upstream binding. We included a truncated ext-pegRNA in PE2-a although it does not create a 2nd nick. (iii) PE2-b: it generates two *cis* nicks, one using the ups-sgRNA, and another by using the ext-pegRNA. PE2-b also does not have the upstream binding. We compared these three PE2 variants with EXPERT, which has the upstream binding, to edit the same *HEK4_2* locus for introducing a 40-bp sequence replacement in HEK293T cells (Supplementary Fig. 2b). All these PE2 variants lacking the upstream binding design, regardless of the design to generate one nick or two *cis* nicks, achieved low efficient edits, ranging from 0.05% to 0.37%, consistent with the knowledge that current PEs are inefficient for long fragment edits. Remarkably, the efficiency achieved by EXPERT is 6.1%, 122.1-fold higher than that by PE2. These results confirmed the essential role of the upstream binding for the EXPERT system.

One consideration with the generation of two nicks in the EXPERT system is whether this will increase the unintended indel rates at the on-target site. Our results in the

above experiments suggest that the presence of two *cis* nicks does not increase the likelihood of indel events in comparison to the one nick system PE2. The indel rate by EXPERT (using ext-pegRNA and ups-sgRNA) at the *HEK4_2* site is 0.28%, comparable to that by PE2 (with 0.2% indels) using the same ups-sgRNA (Supplementary Fig. 2b).

In summary, by introducing an extra upstream guide RNA (ups-sgRNA) to create an additional *cis* nick and an extended pegRNA (ext-pegRNA) that contains an upstream binding sequence, we construct a new PE tool EXPERT.” has been added.

R3.29: Second, and this is a minor point, but could help in interpretation of the data. They use three different colors for inserts, deletions and replacements, but in the graphs to analyze (4b and 4c), the colors are removed – it would be interesting to see if they inserts, indels and replacements are grouped or randomly distributed – simply asking for data points in Fig 4b/c to have the same colors.

Response: Thank you for your suggestion. We have revised figures Fig 4b/c as per your advice. The insertions, deletions, and replacements are now labeled with distinct colors (Corresponding one-to-one with the color in Fig. 4a) to facilitate clearer interpretation of the data points.

Fig. 4 The EXPERT strategy enhances editing efficiency with high product purity and low off-target effects. **a** Frequencies of intended editing and indels introduced by PE2 and EXPERT at multiple loci. Bars represent the mean of n = 3 independent replicates. **b** Statistical analysis of

normalized editing frequencies, setting the frequencies induced by PE2 as 1. n = 19 editing from 3 independent experiments shown in Fig. 4a and supplementary Fig. 8. **c** Statistical analysis of normalized frequencies of indels, setting the frequencies induced by PE2 as 1. n = 19 editing from 3 independent experiments shown in Fig. 4a and supplementary Fig. 8. **d** Frequencies of intended editing and indels introduced by twinPE and EXPERT at different sites. Bars represent the mean of n = 3 independent replicates. **e** Numbers of genome-wide base substitutions. Bars represent the mean of n = 3 independent replicates. **f** Numbers of genome-wide indels. Bars represent the mean of n = 3 independent replicates.

R3.30: Ln 272-282 – suggest supplementary figure to show purity ratios for EMX data

Response: Thank you for your suggestion. We have included the purity ratios for the data in our results section (Supplementary Fig. 14).

Supplementary Fig. 14 Statistical analysis of product purity of PE2max, PE3max, PE4max, PE5max and their corresponding EXPERTmax systems. Bars represent the mean of n = 3 independent replicates.

Specifically we have revised the following:

Lines 409-411: “Furthermore, product purity results demonstrated that EXPERTmax consistently achieved higher purity compared to its corresponding PEmax systems (Supplementary Fig. 14).”has been added.

R3.31: Ln283-284 – I agree, this data does validate the claim that this approach should work with most other prime editors currently available and others that are generated in future. Would be cool to see in the future how EXPERT works with some of the new editors such as PE6, in particular the PE6 version that works best with installing longer edits? Authors could speculate on this or similar ideas in discussion?

Response: Thank you so much for your suggestion. We believe that the EXPERT strategy could theoretically be adapted for use with PE6 and PE7, we have incorporated relevant discussions in the Discussion section.

Specifically we have revised the following:

Lines 514-516: “We believe that the EXPERT strategy, which uses ups-sgRNA and ext-pegRNA, can be readily adopted by other PE systems, such as PE6 (PE6a-g)³⁷ and PE7³⁸ that have been recently reported, as long as they are based on nicking the target DNA.” has been added.

R3.32: Ln 313-314 – given my comments on indels earlier, I think this statement should be modified slightly.

Response: Thank you for your feedback. We have revised our statement accordingly.

Specifically we have revised the following:

Lines 443-444: “we also did not observe significant indel rates.” has been corrected to “the unintended indel rates did not significantly increase compared to PE2max for most edits.”

R3.33: Ln 338 – Figure 5f – suggest minor adjustment to figure 5f such that the epegRNA is in the same place in all four lines of the figure – also, the approximate position of the nicking site for the epegRNA, the ups-gRNA and nicking guide should be shown on exon 4 of the CFTR diagram.

Response: Thank you for your feedback. We have made the following modifications to Figure 5f (now shown in Figure 5g) as per your suggestions:

- i) Ensured that the epegRNA is consistently positioned in the same place across all four lines of the figure.
- ii) Added the approximate position of the nicking site for the epegRNA, the ups-gRNA, and the nicking guide on exon 4 of the CFTR diagram.

Figure 5g. Schematic diagram of complex mutations in *CFTR* exon 4. The intended editing that carried four mutations were performed in HEK293T cells using PE2max, PE3max, EXPERTmax, and EXPERTmax + nicking sgRNA, respectively.

R3.34: Ln345-346 – that statement is too strong for the limited amount of genetic disease data, it should be modified.

Response: Thank you for your feedback. We have revised our statement accordingly.

Specifically we have revised the following:

Lines 474-475: “These findings underscore the potential value of EXPERTmax in translational research of human diseases such as CF.” has been corrected to “These findings suggest the potential value of EXPERTmax in translational research of human diseases such as CF.”

R3.35: Ln350-352 – does the phrase “the 3’ extension binds to the trans strand of the PBS site” make sense – I think it needs modification.

Response: Thank you so much for your feedback. We apologize for our previous inaccurate description. We have rewritten the sentence.

Specifically we have revised the following:

Lines 479-480: “The 3’ extension of the ext-pegRNA binds to the trans strand that of the PBS site that is bound by the ups-sgRNA.” has been corrected to “The 3’ extension of the ext-pegRNA binds to the region targeted by the ups-sgRNA.”

R3.36: Ln 355 – “region” is better than “sides”

Response: Thank you for your suggestion. We have revised it accordingly.

Specifically we have revised the following:

Lines 483-484: “sides” has been changed to “regions”.

R3.37: Ln357 – a 122-fold increase is claimed here and in the abstract, but I don’t see a justification for it – can it be provided/clarified or removed.

Response: We apologize for the oversight. As we stated in response to **R3.28**. We have now provided clarification in the revised manuscript and re-written this sentence.

Specifically we have revised the following:

Lines 119-156: “Both *cis* nicks and the upstream binding are essential for the differential editing capacity of the EXPERT system vs the canonical PEs. We first confirmed the role of the two *cis* nicks in the EXPERT system, by comparing it with the two EXPERT variants, EXPERT-a and EXPERT-b, that only generate one nick by using only the ups-sgRNA or only the ext-pegRNA, respectively (Supplementary Fig. 2a). We used them to edit the *HEK4_2* locus for introducing a 40-bp sequence replacement in HEK293T cells (Supplementary Fig. 2b). As expected, EXPERT (with two *cis* nicks) achieved 6.1% efficiency, in contrast to the 2.84% and below 0.1% efficiencies achieved by EXPERT-a and EXPERT-b (the single nick systems), respectively. Mechanistically, we speculate that this is because EXPERT generates two *cis* nicks that enhances the detachment of the original single-stranded DNA

fragment from the genome, thereby promoting subsequent processes (Supplementary Fig. 2c). These results confirmed the importance of the two *cis* nicks for the EXPERT system.

We next worked to confirm the role of the upstream binding. We included several PE2 variants in the experiment (Supplementary Fig. 2a). (i) PE2: it generates one nick by using the ups-sgRNA as its pegRNA. It does not have the upstream binding. (ii) PE2-a: it generates one nick by using the ups-sgRNA as its pegRNA. It does not have the upstream binding. We included a truncated ext-pegRNA in PE2-a although it does not create a 2nd nick. (iii) PE2-b: it generates two *cis* nicks, one using the ups-sgRNA, and another by using the ext-pegRNA. PE2-b also does not have the upstream binding. We compared these three PE2 variants with EXPERT, which has the upstream binding, to edit the same *HEK4_2* locus for introducing a 40-bp sequence replacement in HEK293T cells (Supplementary Fig. 2b). All these PE2 variants lacking the upstream binding design, regardless of the design to generate one nick or two *cis* nicks, achieved low efficient edits, ranging from 0.05% to 0.37%, consistent with the knowledge that current PEs are inefficient for long fragment edits. Remarkably, the efficiency achieved by EXPERT is 6.1%, 122.1-fold higher than that by PE2. These results confirmed the essential role of the upstream binding for the EXPERT system.

One consideration with the generation of two nicks in the EXPERT system is whether this will increase the unintended indel rates at the on-target site. Our results in the above experiments suggest that the presence of two *cis* nicks does not increase the likelihood of indel events in comparison to the one nick system PE2. The indel rate by EXPERT (using ext-pegRNA and ups-sgRNA) at the *HEK4_2* site is 0.28%, comparable to that by PE2 (with 0.2% indels) using the same ups-sgRNA (Supplementary Fig. 2b).

In summary, by introducing an extra upstream guide RNA (ups-sgRNA) to create an additional *cis* nick and an extended pegRNA (ext-pegRNA) that contains an upstream binding sequence, we construct a new PE tool EXPERT.” has been added.

Lines 485-489: “Efficiency wise, the EXPERT strategy significantly improves the efficiency, up to 122-fold improvement over that by PE2. Safety wise, the EXPERT strategy does not increase the risks of off-target editing and on-target undesirable indel generation.” has been corrected to “Efficiency wise, the EXPERT strategy significantly improves the efficiency, showing an average improvement of 3.12-fold (up to 122.1 times higher) compared to PE2. Safety wise, the EXPERT strategy does not increase the risks of off-target editing and generally results in low on-target indel generation relative to PE2.”

R3.38: Ln 358 – should that sentence end with “relative to PE2”?

Response: Thank you so much for your suggestion. We have revised the sentence accordingly.

Specifically we have revised the following:

Lines 488-489: “relative to PE2” has been added.

R3.39: Ln 363 – I don’t think 2.9% indels is extremely low – also, 2.9% editing seems to conflict with statement on ln357/358 about low indels?

Response: Thank you for your feedback. We have revised this sentence. Additionally, we have included further discussion about indels in the Discussion section.

Specifically we have revised the following:

Lines 492-494: “In fact, the EXPERT-associated indel rates observed in the experiments are extremely low, ranging from 0.1% to 2.9%, comparable to or even lower than those generated by PE2.” has been corrected to “In fact, the EXPERT-associated indel rates observed in the “PE2 vs EXPERT” experiments are in the range from 0.1% to 4.2%, comparable to or lower than those generated by PE2.”

R3.40: Ln 391-393 good point.

Response: Thank you for your comments.

Point-by-point response to reviewers' comments

Reviewer #1 (R1):

Xiong et al have provided additional data to strengthen this manuscript. There are a few issues that still remain and need to be addressed.

Response: Thank for your valuable feedback. Please see our responses to your specific comments below.

R1.1: The authors did not show EXPERT's performance on single-base substitutions in response 1.1.1.

Response: Thank you. In response, we have added the single-base substitutions data to Response 1.1.1 (Supplementary Fig. 5a).

Please also see our responses to R1.3 and R1.4.

Supplementary Fig. 5a Frequencies of intended small edits and indels were performed by EXPERT. Bars represent the mean of n = 3 independent replicates.

Specifically we have revised the following:

Lines 261-266: “The results showed that EXPERT can perform all these types of small edits in the upstream region of the ext-pegRNA nick (Supplementary Fig. 5). Again, introducing mismatches significantly improved the editing efficiency, in line with our speculation that introduced mismatches reduces the homology between the ext-pegRNA and the genomic DNA 3' Flap, thereby preventing their hybridization (Fig. 2c).” has been revised.

R1.2: The number of twinPE pegRNA pairs screened was not specified. Dual pegRNA methods, including twinPE, require screening and optimization for high efficiency and

purity at the genomic locus. Please include this information for a fair comparison between twinPE and EXPERT, which also requires optimization.

Response: Thank you for your professional comments. We apologize for not showing our screening of twinPE pegRNA pairs in the previous manuscript. In response, we have added these in the revision. Briefly, a series of twinPE pegRNA pairs for these different loci were evaluated (Supplementary Fig. 11). Then, the best-performing twinPE pegRNA pairs (with a precise editing efficiency of at least 2% and the highest purity) for each locus were selected and used for comparison with EXPERT (Fig. 4d).

Supplementary Fig. 11 Frequencies of intended editing and indels, and the product purity introduced by different twinPE pegRNA pairs at endogenous sites. The red word represents the selected twinPE with the best performance (with precise editing efficiency of at least 2% and the highest purity). Bars represent the mean of n = 3 independent replicates.

Fig. 5d Frequencies of intended editing and indels introduced by twinPE and EXPERT at different sites. Bars represent the mean of n = 3 independent replicates.

Specifically we have revised the following:

Lines 355-356: “The twinPE pegRNA pairs used for each locus were selected after a pre-screening process (Supplementary Fig. 11).” has been added.

Lines 578-582: “Pre-screening of twinPE pegRNA pairs

A series of twinPE pegRNA pairs for these different loci were evaluated, and shown in Supplementary Fig. 11. The best-performing twinPE pegRNA pairs, with a precise editing efficiency of at least 2% and the highest purity, for each locus were selected and used for comparison with EXPERT.” has been added.

R1.3: In R1.7, the edits primarily involve large sequence replacements, deletions, or insertions. Since prime editing (as per Anzalone et al., Nature, 2019) is highly efficient at single-base substitutions and small indels, the authors should include single-base substitution edits using PE3 (1 pegRNA + 1 nicking sgRNA) and compare with EXPERT (1 pegRNA + 1 nicking sgRNA).

Response: Thank for your valuable feedback. In response, we have compared the single-base substitutions efficiency of PE3 and EXPERT. As expected, the editing efficiency of EXPERT for small edits is lower than that of PE3 (Supplementary Fig. 5b). This is because EXPERT was originally designed to address the issue of low efficiency in large fragment edits, a problem commonly associated with existing prime editing variants. Interestingly, however, the unintended on-target indel rates were lower with EXPERT than with PE3. Nevertheless, we have added these results in the revision, and added a discussion in the revision pointing out that EXPERT is suitable for large fragment edits, while base editors and PE3 are better choices for single-base substitutions and small indels. We should carefully choose the most appropriate tool based on the specific situation.

Please also see our responses to R1.1 and R1.4.

Supplementary Fig. 5b Frequencies of intended small edits and indels were performed by EXPERT and PE3. Three mismatches were included in EXPERT. Bars represent the mean of $n = 3$ independent replicates.

Specifically we have revised the following:

Lines 266-268: “Nevertheless, it is noted the efficiency of these small edits achieved by EXPERT are generally lower than that achieved by PE3 (Supplementary Fig. 5).” has been added.

Lines 535-538: “We also want to point out that EXPERT is suitable for large fragment edits (large sequence replacements, deletions, or insertions) and further improvements are needed for small edits. Until then, other tools such as base editors and PE3 are good choices for single-base substitutions and small indels.” has been added.

R1.4: The revised manuscript lacks sufficient evidence to support the claim that “EXPERT enhances editing efficiency of all types.” Specifically, single-base substitution comparisons were made between PE2 (using one pegRNA) and EXPERT, which involved additional optimization. PE3 and PE5, with similar optimization, should be used to substantiate this claim. Authors should revise their statement to accurately describe the advancement of EXPERT to the field.

Response: Thank you for your insightful suggestion. We have revised our statement based on your suggestions and our results. We agree that EXPERT is suitable for large fragment edits (large sequence replacements, deletions, or insertions) and further improvements are needed for small edits. In response, we have revised the statement throughout the manuscript. We hope these revisions address your concern and provide a more accurate description of the advancements presented by EXPERT.

Please also see our responses to R1.1 and R1.3.

Specifically we have revised the following:

Lines 2-3: The title of the manuscript has been revised to “EXPERT system Expands the Editing Region of Prime Editing and Enhances its Efficiency for Large Fragment Edits”

Line 42: “of all types” has been revised to “for large fragment edits”.

Lines 535-538: “We also want to point out that EXPERT is suitable for large fragment edits (large sequence replacements, deletions, or insertions) and further improvements are needed for small edits. Until then, other tools such as base editors and PE3 are good choices for single-base substitutions and small indels.” has been added.

R1.5: Cellular repair machinery varies among cell types (e.g., transformed vs. primary cells). Data from non-transformed or therapeutically relevant cells (e.g., T cells, iPSCs, primary fibroblasts) are needed to evaluate EXPERT's broader potential in life sciences.

Response: Thank you for your valuable suggestion. We apologize for not addressing

this point more clearly in our previous submission. To evaluate the broader applicability of EXPERT, we conducted experiments in both Jurkat cells (T cells) (Fig. 5c) and PFF cells (pig fetal fibroblasts) (Fig. 5f). The results demonstrated that EXPERTmax significantly enhanced editing efficiency and product purity compared to PE2max in different cell types.

In response to your suggestion, and to further assess the potential of EXPERT in therapeutically relevant cell types, we have added experiments in human primary fibroblasts (HFL1). The results showed that the efficiency of EXPERTmax is 1.5 to 25.6-fold higher than that of PE2max, while the unintended indel rates remain either lower than or comparable to those of PE2max (Supplementary Fig. 18). We have now incorporated these results into the Results section to further support the broader applicability of EXPERT.

Fig. 5c Frequencies of intended editing and indels introduced by PE2max and EXPERTmax in human leukemia T lymphocyte Jurkat cells. Bars represent the mean of n = 3 independent replicates.

Fig. 5f Frequencies of intended editing and indels, and the product purity introduced by PE2max and EXPERTmax at endogenous sites in pig fetal fibroblast (PFF) cells. Bars represent the mean of n = 3 independent replicates.

Supplementary Fig. 18 Frequencies of intended editing and indels introduced by PE2max and EXPERTmax at endogenous sites in human fetal lung fibroblast (HFL1) cells. Bars represent the mean of $n = 3$ independent replicates.

Specifically we have revised the following:

Lines 431-434: “In HFL1 cells, the efficiency of EXPERTmax is 1.5 to 25.6-fold higher than that of PE2max, while the unintended indel rates remain either lower than or comparable to those of PE2max (Supplementary Fig. 18).” has been added.

Reviewer #2 (R2):

Thank you for carefully considering the suggestions. The responses were complete.

Response: Thank for your positive feedback, recognition of this work, and support in the publication of this study.

Reviewer #3 (R3):

All questions addressed thoroughly, no further comments.

Response: Thank for your positive feedback, recognition of this work, and support in the publication of this study.

Point-by-point response to reviewers' comments

Reviewer #1 (R1):

Thank you for conducting additional experiments to address my comments and strengthen the manuscript. I suggest that the authors exercise caution when drawing conclusions from their data. While the manuscript demonstrates clear strengths—such as the ability of EXPERT to target the upstream region of the ext-pegRNA, offering opportunities to improve efficiency and reduce indels for certain edits. Some conclusions in the rebuttal and manuscript text are not stated properly. I recommend that the authors revise the manuscript main text accordingly. Below, I provide three examples for improvement:

Response: Thank for your valuable feedback. We have revised the descriptions and conclusions in the manuscript accordingly.

R1.1: In R1.3 (Supplementary Figure 5b):

You stated, "Interestingly, however, the unintended on-target indel rates were lower with EXPERT than with PE3." Readers may not fully appreciate this conclusion given that the on-target editing efficiency for EXPERT is very low, 4- to 12-fold less efficient than PE3. Moreover, the fold improvement in indel reduction with EXPERT is only 2- to 3-fold better than PE3. If you wish to discuss product purity, I recommend using the editing specificity ratio, which is commonly reported in genome editing literature.

Response: Thank you for your professional comments. We apologize for the unprofessional description in previous Response. We have now revised Supplementary Figure 5b to better reflect the data, which demonstrate that EXPERT has a lower editing efficiency on the small edits compared to PE3.

Here, we have also presented the product purity by using the editing specificity ratio (intended edits: indels ratio), which is commonly reported in genome editing literature (Chen Peter J. et al. 2021). As expected, the product purity of EXPERT is lower than that of PE3, indicating that EXPERT is not suitable for small edits (Supplementary Fig. 5b). Further optimization is needed for EXPERT to be used for small edits.

Supplementary Fig. 5b Frequencies of intended small edits, indels, and product purity introduced by EXPERT and PE3. Three mismatches were included in EXPERT. Bars represent

the mean of $n = 3$ independent replicates. All sequencing data is collected from transfection-positive cells.

Specifically we have revised the following:

Lines 266-269: “Nevertheless, it is noted that despite a lower unintended on-target indel rates, the efficiency as well as the product purity of the small edits achieved by EXPERT is generally lower than that achieved by PE3 (Supplementary Fig. 5b).” has been revised.

R1.2: Regarding the twin prime editing pegRNA screening:

In Anzalone et al. NBT 2022, the authors screened multiple pairs of pegRNAs, PBS lengths, and various combinations of pegRNAs. Liu Lab conducted a more comprehensive evaluation of pegRNAs to select the best twin prime pegRNA pairs that demonstrate low indels and high efficiency for downstream applications. I understand that performing pegRNA screening at the same scale and making thorough comparisons is challenging in this manuscript. However, it would be more informative to acknowledge the specific parameters evaluated in your twin prime editing pegRNA screen and state a more comprehensive comparison would be recommended for further investigation.

Response: Thank for your valuable feedback. We agree that although a pre-screening process for twinPE pegRNA pairs has been conducted, a more thorough screening (including additional pegRNA pairs, varying PBS lengths, and different combinations) will be essential in the future to enable a more comprehensive comparison of the performance between EXPERT and twinPE. We have added relevant statements to the main text to reflect this. We have also included the relevant description in our twinPE pegRNA screening.

Specifically we have revised the following:

Lines 363-367: “It should be noted that although a pre-screening process for twinPE pegRNA pairs has been conducted, a more thorough screening (including additional pegRNA pairs, varying PBS lengths, and different combinations) will be essential in the future to enable a more comprehensive comparison of the performance between EXPERT and twinPE.” has been added.

Lines 585-588: “Specifically, for each locus, we tested various combinations of pegRNA pairs, with the PBS of each pegRNA designed using the online tool pegFinder³⁶. The detailed sequence information for these pegRNAs is provided in Supplementary Table 1.” has been added.

R1.3: Additional cell line data and method clarification:

The additional data from transformed cell lines and primary cell HFL1 experiments

support the applicability of EXPERT in other cell types. However, the figure legend should specify whether the % editing shown in the bar graph represents FACS-enriched, transfected cells. If so, what is the non-selected, bulk cell editing efficiency? Please update the corresponding Methods section, as the current description of the FACS selection protocol is insufficient for replication.

Response: Thank for your valuable feedback. All data in this study were collected through sequencing of transfection-positive cells. HEK293T, HeLa, N2a and PFF cells were enriched by puromycin selection, while K562, Jurkat, and HFL1 cells were enriched by flow cytometric sorting of GFP-positive cells (transfection-positive cells).

For the concern about “the non-selected, bulk cell editing efficiency”. In most prime editing studies, sorting/drug selection are used to enrich the successfully transfected cells (Liu, Y. et al. 2021, Zhuang Y. et al. 2022, and A. Sousa Alexander et al. 2024). Therefore in the present study, we utilized the same strategy and evaluated editing efficiencies in enriched cells. We did not evaluate the non-selected bulk editing efficiency.

In response, we have revised all the figure legends and added the relevant description to the Methods section.

Supplementary Fig. 18 Frequencies of intended edits and indels introduced by PE2max and EXPERTmax at endogenous sites in human fetal lung fibroblast (HFL1) cells. Bars represent the mean of n = 3 independent replicates. All sequencing data is collected from transfection-positive cells.

Specifically we have revised the following:

Lines 609-617: “All data in this study were collected through sequencing of transfection-positive cells. K562, Jurkat, and HFL1 cells were enriched with positive cells using flow cytometry sorting. Specifically, At 96 h post-transfection, GFP-positive cells (transfection-positive cells) were FACS-enriched (BD Aria™ III) for subsequent experiments. HEK293T, HeLa, N2a, and PFF cells were enriched by adding puromycin (InvivoGen) to the medium at the final concentration of 2 µg/mL, 24 h post-transfection. At 96 h post-transfection, transfection-positive cells were collected and lysed with freshly prepared lysis buffer and incubated at 65 °C for 40 min, followed by incubation

at 95 °C for 15 min.” has been revised.

Lines 622-624: “For Flow cytometry sorting, K562, Jurkat, and HFL1 cells were collected after PBS washing, resuspended in PBS, and sorted using a BD Aria™ III flow cytometer, followed by enrichment of GFP-positive cells.” has been revised.

I recommend publication of this manuscript once the authors address the comments outlined above and carefully revise manuscript main text.

Response: Thank for your positive feedback, recognition of this work, and support in the publication of this study. Based on your suggestions, we have revised the relevant descriptions and conclusions in the revision. Additionally, we have carefully revised the main text of the manuscript.